



# Characterization of aerosol over the Eastern Mediterranean by polarization sensitive Raman lidar measurements during A-LIFE – aerosol type classification and type separation

Silke Groß[1], Volker Freudenthaler[2], Moritz Haarig[3], Albert Ansmann[3], Carlos Toledano[4], David Mateos[4],
Petra Seibert[5, 6], Rodanthi-Elisavet Mamouri[7], Argyro Nisantzi[7], Josef Gasteiger[8,a], Maximilian Dollner[8],
Anne Tipka[5], Manuel Schöberl[8], Marilena Teri[8], Bernadett Weinzierl[8]

[1]Deutsches Zentrum für Luft- und Raumfahrt (DLR) e.V., Institut für Physik der Atmosphäre, Oberpfaffenhofen, Germany
[2]Ludwig-Maximilians-Universität, Meteorologisches Institut, München, Germany
[3]Leibniz-Institut für Troposphärenforschung (TROPOS), Leipzig, Germany
[4]Universidad de Valladolid, Valladolid, Spain
[5]Universität für Bodenkultur Wien, Institute of Meteorology and Climatology, Wien, Austria
[6]Universität Wien, Institut für Meteorologie und Geophysik, Wien, Austria
[7]Eratosthenes Center of Excellence, Limassol, Cyprus
[8]Universität Wien, Aerosol Physics and Environmental Physics, Wien, Austria
[a]Now at: Hamtec Consulting GmbH at EUMETSAT, Darmstadt, Germany

*Correspondence to*: Silke Groß (silke.gross@dlr.de)

**Abstract.** Aerosols are key players in the Earth's climate system with mineral dust being one major component of the atmospheric aerosol load. While former campaigns focused on investigating the properties and effects of rather pure mineral dust layers, the A-LIFE (**A**bsorbing aerosol layers in a changing climate: aging, **life**time and dynamics) campaign in April 2017 aimed to characterize dust in complex aerosol mixtures. In this study we present ground-based lidar measurements that were performed at Limassol, Cyprus, in April 2017. During our measurement period, the measurement site was affected by complex mixtures of dust from different sources and pollution aerosols from local sources as well as long-range transported. We found mean values of the particle linear depolarization ratio and extinction-to-backscatter ratio (lidar ratio) of $0.27 \pm 0.02$ and 41 sr $\pm$ 5 sr at 355 nm and of $0.30 \pm 0.02$ and 39 sr $\pm$ 5 sr at 532 nm for Arabian dust, and of $0.27 \pm 0.02$ and 55 sr $\pm$ 8 sr at 355 nm and of $0.28 \pm 0.02$ and 53 sr $\pm$ 7 sr at 532 nm for Saharan dust. The values found for pollution aerosols of the particle linear depolarization ratio and the lidar ratio are $0.05 \pm 0.02$ at 355 nm and $0.04 \pm 0.02$ at 532 nm, and 65 sr $\pm$ 12 sr at 355 nm and 60 sr $\pm$ 16 sr at 532 nm, respectively. We use our measurements for aerosol typing and compare that to aerosol typing from sun photometer data, in-situ measurements and trajectory analysis. The different methods agree well for the derived aerosol type, but looking at the derived dust mass concentration from different methods, the trajectory analysis frequently underestimate high dust concentration that were found in major mineral dust events.



## 1 Introduction

Aerosol particles are omnipresent and can affect the Earth's atmosphere in different ways: they directly interact with incoming solar or outgoing terrestrial radiation by scattering and absorption, and they indirectly affect the formation and properties of clouds by acting as cloud condensation nuclei or ice nuclei. Additionally, they can also change the atmosphere's temperature and stability profile. Up to now, aerosols are contributing to the largest uncertainties in estimating changes of the Earth's climate system (Forster et al., 2007; Boucher et al., 2013; Bender 2020). One reason is the aerosol's strong temporal and spatial variability. Furthermore, the sign and the magnitude of their radiative impact strongly depends on the microphysical and chemical properties of the aerosol particles as well as on their vertical distribution. During their lifetime and transport the aerosol particles are exposed to transformation processes such as particle aging or mixing process. These can change the optical, microphysical properties and the ability of the aerosol to act as cloud condensation nuclei. In addition, the aerosol properties can change due to mixing of different types.

In-situ measurements directly measure the microphysical (e.g. Kaaden et al., 2009; Weinzierl et al., 2011) and chemical particle properties (Kandler et al., 2009), and can be used to derive the aerosol particle's ability to act as cloud condensation nuclei (e.g. Kumar et al., 2011; Haarig et al., 2019). These in-situ measurements, however, are strongly limited in space and time. To estimate the radiative and cloud influencing properties of aerosol layers from remote sensing measurements, further information and/or aerosol classification schemes are crucial, as different particle types interact differently with incoming and outgoing radiation and have a different impact on cloud formation and properties. Passive remote sensing measurements with sun photometer provide column integrated values of aerosol properties and thus can only give a column integrated typing (Toledano et al., 2011). Lidar measurements provide profile information of the aerosol and cloud structure. Polarization sensitive Raman or High Spectral Resolution Lidar (HSRL) systems provide height resolved information of intensive optical properties (i.e. lidar ratio and particle linear depolarization ratio) that can be used for aerosol typing (Burton et al., 2012; Groß et al., 2011b, 2015b; Nicolae et al., 2018). In a next step the aerosol layers can be linked to typical microphysical properties for the derived aerosol type (Groß et al., 2013a, Wandinger et al., 2023, Floutsi et al., 2023) to calculate the radiative effect of the aerosol layer (Gutleben et al., 2020, 2021) or their ability to act as cloud condensation nuclei (Ansmann et al., 2019) or ice nuclei (Mamouri and Ansmann 2016; Marinou et al., 2019). However, aerosol classification schemes are limited in the considered aerosol types. Additionally, different classification schemes rely on different measurement properties, and thus might differ in the derived results e.g. for aerosol mixtures. It is thus necessary to constantly further develop aerosol typing schemes and to re-evaluate them by comparison of classification schemes based on different measurement methods.

Mineral dust is a main contributor to the atmospheric aerosol load (Haywood and Boucher, 2010). Mineral dust scatters and absorbs the incoming and outgoing radiation, but the magnitude and sign of the dust radiative forcing is still not fully clear (e.g. Kok et al., 2018; Adebiyi et al., 2023). It strongly depends on the microphysical properties and chemical composition of the dust particles, which differ for dust particles from different sources (Kandler et al., 2009; Lieke et al., 2011). Dust



microphysics and chemical composition have an impact on their optical properties (e.g. Groß et al., 2011b; Schuster et al.,
2012; Nisantzi et al., 2015). In addition, the irregular shape of the dust particles causes difficulties in the modelling of the dust
radiative effects (Gasteiger et al., 2011, Saito et al., 2021). To increase our knowledge, a large number of studies were
performed. For example, lidar measurement in the framework of the European Aerosol Research Lidar Network (EARLINET;
Papalardo et al., 2014) at different measurement sites in Europe were analysed to study mineral dust transport towards Southern
Europe (e.g. Cachorro et al., 2008; Bravo-Aranda et al., 2015; Mona et al., 2014; Navas-Guzman et al., 2013), Central Europe
(e.g. Ansmann et al., 2003; Wiegner et al., 2012, Haarig et al., 2022), Eastern Europe (Binietoglou et al., 2015; Talianu et al.,
2007) and the Mediterranean (Amiridis et al., 2009; Papayannis et al., 2009; Mamouri et al., 2013; Soupiona et al., 2020). In
addition, several field experiments have taken place to study mineral dust at different locations and stages of lifetime (e.g.
SHADE – Tanré et al., 2003; PRIDE – Reid et al., 2003; FENNEC – Ryder et al., 2013). The most comprehensive field
experiment to study mineral dust was the Saharan Mineral Dust experiment (SAMUM – Ansmann et al., 2011) which was
followed by the Saharan Aerosol Long-range Transport and Aerosol-Cloud-Interaction Experiment (SALTRACE – Weinzierl
et al., 2017). SAMUM and SALTRACE were designed as closure studies, combining airborne and ground-based in-situ, lidar
and radiation measurements together with modelling efforts. The optical, microphysical, chemical and radiative properties of
Saharan mineral dust were studied close to the source region, at the beginning and after long-range transport towards the
Caribbean.

With a similar concept, the A-LIFE (**A**bsorbing aerosol layers in a changing climate: aging, **life**time and dynamics) field
experiment was performed in Cyprus in April 2017 (https://a-life.at). The Eastern Mediterranean is a hotspot for different types
of aerosols. Mineral dust from Africa, Asia and the Arabian Peninsula are frequently transported towards Cyprus. In addition,
the region is affected by biomass burning aerosol from forest fires and by local and transported pollution. This makes this
region an ideal location to study mineral dust from different source regions and to investigate the impact of aging and mixing.
In this study, we aim to investigate the optical properties of mineral dust from source regions in the Sahara and from the
Arabian Peninsula, to study differences of the different dust types and other absorbing aerosol, and to determine how these
results impact aerosol classification schemes. For this it is important to compare the different methods for aerosol type
classification and their ability to retrieve dust mass concentration. In Section 2, we present the used measurements and methods.
Section 3 gives the results of this study focusing on the characterization of the general measurement situation, the optical
properties of the observed aerosol types, and an aerosol typing from different methods. In Section 4, we discuss the agreement
of the different typing methods as well as the derived dust contribution. Section 5 concludes this work.



## 2 Methodology

### 2.1 A-LIFE field experiment

For the analysis presented in this study we use ground-based lidar measurement that were performed during the A-LIFE field
experiment as part of the ERC funded project A-LIFE (https://a-life.at; Weinzierl et al., in prep.). The experiment was designed
as a closure experiment combining airborne remote sensing and in-situ measurements onboard the DLR Falcon together with
ground-based observations, long-term observations and modelling efforts. Measurements were performed in Cyprus in April
2017. The DLR Falcon was based at the airport at Paphos, where also ground-based in-situ measurements were performed.
The lidar measurements with the POLIS and Polly[XT] together with sun-photometer measurements were performed at Limassol.
A detailed description of the experiment is given in Weinzierl et al., (in prep.).

### 2.2 POLIS lidar system

POLIS (portable lidar system) is a small, six-channel, dual-wavelength polarization sensitive Raman lidar system which was
developed and built at the Ludwig-Maximilian-Universität München (Groß et al., 2015a). POLIS measures simultaneously the
co- and cross-polarized light at 355 nm and 532 nm. During night-time additional measurements of the N2-Raman shifted
wavelengths at 387 nm and 607 nm are performed. The full overlap of the small lidar system is adjustable from about 70 m
and was about 200 m during the campaign, allowing high accurate measurements within the boundary layer. The measured
raw data has a resolution of 3.75 m in range and typically 10 s in time. Additionally, a 25-bin sliding average, i.e. ~94 m, is
used to reduce signal noise. For the night-time Raman measurements, the data is typically averaged over 1.5 – 2 h in time and
with a 151-bin sliding average, i.e. ~566 m. The Raman approach described by Ansmann et al. (1990; 1992) is used to directly
retrieve the extinction coefficient and backscatter coefficient and thus the extinction-to-backscatter ratio (lidar ratio). The lidar
ratio is then used in the Fernald-Klett algorithm (Fernald, 1984, Klett, 1985) to retrieve the particle linear depolarization ratio
(PLDR) with higher spatial resolution (sliding average of 25 or 51 bins) and for the day-time measurements. For the day-time
analysis, the measurements are averaged over 1 hour around coordinated in-situ measurements onboard the DLR Falcon
aircraft. For the analysis of the PLDR the high accurate Δ90 calibration method (Freudenthaler et al., 2009, Freudenthaler et
al., 2016a; Freudenthaler, 2016b) was used. The uncertainties of retrieved properties were calculated following the procedure
described by Freudenthaler et al. (2009) and Groß et al. (2011a)

### 2.3 Polly[XT]

The POrtabLe Lidar sYstem (with eXTended capabilities) Polly[XT] of TROPOS is described in Engelmann et al (2016). The
latest status of data analysis can be found in Baars et al. (2016), Hofer et al. (2017), and Ohneiser et al. (2020). The capabilities
of the multiwavelength polarization-sensitive Raman lidar are similar to the POLIS described above. The Polly[XT] instrument





was continuously operated over the one-month A-LIFE campaign. The same quality standards regarding data and uncertainty analysis as in the case of the POLIS data analysis are applied.

While POLIS measures the co- and cross-polarized backscatter signal component, Polly$^{XT}$ measures the total and cross-polarized backscatter signal component. While it is not in the focus of this paper to discuss minor differences resulting from

the differences in the system setup, it is still worth to intercompare the measurements and the resulting classification.

## 2.4 Aerosol typing and aerosol type separation based on lidar measurements

To determine the aerosol type in case of rather pure aerosol situations, i.e. no mixture of different aerosol types, we used the retrieved PLDR and lidar ratio based on the classification schemes described by Groß et al. (2013a, 2015b). To describe the contribution of different aerosol types in an aerosol mixture, we use the PLDR and the backscatter coefficient measured at 532

nm following the procedure described by Tesche et al. (2009a) and Groß et al. (2011c, 2016). We assume a two-type mixture of mineral dust and pollution, which is in good agreement with the coordinated in-situ measurements (see Section 4). We follow the procedure described by Tesche et al. (2009a) and Groß et al. (2011c, 2016) to derive the dust and non-dust backscatter coefficient and extinction coefficient. As input for the type separation at 532 nm we use PLDR=0.3 for dust aerosols and PLDR=0.03 for non-dust aerosols according to findings of pure mineral dust (e.g. Freudenthaler et al., 2009, Tesche et al.,

2009b, Groß et al., 2011b, Groß et al., 2015a, and findings of this study) and for anthropogenic pollution (e.g. Groß et al., 2013b, Hofer et al., 2017). The other input for the type separation at 532 nm are a lidar ratio of 55 sr for Saharan mineral dust (e.g. Tesche et al., 2009b, Groß et al., 2013a), of 45 sr for Arabian dust (Mamouri et al., 2013; Nisantzi et al., 2015, this study), and of 70 sr for anthropogenic pollution (Groß et al., 2013b, and this study).

## 2.5 Conversion to volume and mass concentration

The extinction to volume conversion factor of mineral dust from different source regions (e.g. North Africa and Middle East) was intensively studied by Mamouri and Ansmann (2017) and Ansmann et al. (2019) using AERONET (Aerosol Robotic Network; Holben et al., 1998) measurements and inversion products. They found a mean extinction to volume conversion factor for dust of $0.65 \times 10^{-6}$ m. The dust mass concentration is then calculated using the dust volume concentration and multiplying it with the particle density, which we assume to be 2.5 g cm$^{-3}$ according to previous studies (e.g. Wagner et al.,

2009; Groß et al., 2016). For the conversion from extinction to volume of pollution aerosols we use a conversion factor of $0.41 \times 10^{-6}$ m and a particle density of 1.5 g cm$^{-3}$ as proposed by Mamouri and Ansmann (2017) from Limassol AERONET data.

## 2.6 AERONET sun-photometer

AERONET measurements were performed on the roof-top of the Cyprus University for Technology in Limassol about 200 m apart from the lidar site (site name: CUT-TEPAK). Direct sun observations provide the aerosol optical depth (AOD) at eight

spectral channels at wavelengths between 340 nm and 1640 nm. Additionally, optical and microphysical aerosol properties are



derived from the multi-angle and multi-spectral measurement of sky radiance (almucantar and hybrid scan geometries every hour). For details on the instrument calibration and data products see Holben et al., (1998), Dubovik and King (2000), and Dubovik et al. (2006). For this study we use the AOD measurements at 340 nm, 500 nm and 1020 nm as well as the retrieved coarse and fine mode AOD at 500 nm and the Ångström exponent (440nm-870nm and 380nm-500nm) from AERONET

version 3 database (Giles et al., 2019). Further information on the sun-photometer measurements during A-LIFE is provided by Mateos et al. (2024).

**2.7 Aerosol In-situ measurements and A-LIFE in-situ aerosol classification scheme**

For A-LIFE, the DLR research aircraft Falcon was equipped with comprehensive aerosol in-situ instrumentation, a wind lidar and sensors for measuring meteorological parameters. The particle size distribution was measured with a combination of

condensation nuclei counters, optical spectrometers, an optical array probe covering the particle diameter range from 10 nm to 930 µm (Weinzierl et al., in prep.; Schöberl et al., in prep.). The particle scattering coefficients were determined at three different wavelengths ($\lambda = 450, 525, 635$ nm) with a polar nephelometer (Teri et al., 2022, 2024). The absorption coefficient was measured using a tri-color absorption photometer at multiple wavelengths ($\lambda = 465, 520, 640$ nm), while the black carbon mass concentration was determined with a single-particle soot photometer (Teri et al., 2024).

An algorithm was developed to classify the airborne aerosol data into 12 aerosol types consisting of four main aerosol types (Saharan dust, Arabian dust, mixtures with and without coarse mode). Each of the four main aerosol types was further separated into three sub-classes (pure, moderately-polluted and polluted) based on the relative contribution of pollution (Weinzierl et al., in prep). The classification scheme is based on in-situ measurements of coarse mode particle number concentration and refractory black carbon mass concentration. Furthermore, it uses information about the dust source region from the Lagrangian

transport and dispersion model FLEXPART version 8.2 (Stohl et al., 1998, Seibert and Frank, 2004). FLEXPART was driven by meteorological data from the European Centre for Medium-Range Weather Forecasts (ECMWF). Coupled with emission data from the Copernicus Atmospheric Monitoring Service, it provides quantitative information about observed aerosol types and their origins. Here we use the results of the in-situ classification for 23 periods of co-located measurements when the Falcon research aircraft was overflying the ground-based lidar site (see Table 2). Details about the aerosol classification scheme

and its validation are given in Weinzierl et al. (in prep.).

**2.8 Atmospheric transport simulations with FLEXPART**

Backward atmospheric transport simulations were carried out along the flight paths with the Lagrangian particle disperson model FLEXPART (Stohl et al., 1998; Seibert and Frank, 2004; Stohl et al., 2005). Source-receptor relationships obtained

were then combined with emission inventory data from the Copernicus Atmospheric Monitoring Service to obtain simulated





mass concentrations of different species (dust, black carbon, organic matter, sulfate, seasalt). Furthermore, the contributions per species were split into source regions. Based on this output, each 1-min section of the flight tracks was assigned an aerosol type.

## 2.9 HYSPLIT

To identify the source regions and transport ways of the observed aerosol layers by ground-based lidar and sun-photometer measurements, we use back-trajectory calculations. The trajectories were calculated with the Hybrid Single Particle LaGRANGIAN Integrated Trajectory (HYSPLIT) model (Draxler and Rolph, 2012) and reanalysis meteorological data. Start time and height of the trajectories were chosen according to the analyzed lidar measurement time periods and the height ranges of the presumed aerosol layer. The duration of the backward trajectories is 48 h.

**3 Results**

## 3.1 General measurement situation

During the A-LIFE field experiment we observed a high variability of aerosol types transported to our measurement site in Limassol, Cyprus. Satellite measurements as well as trajectory calculations indicated that the main contributing aerosol types were Arabian dust, Saharan dust and pollution/smoke aerosols. The different aerosol events showed a variety of aerosol layer

heights and aerosol optical properties. Frequently, mixtures of different aerosol types or different aerosol types at different height levels were found (https://a-life.at). Figure 1 gives an overview of the measurement situation, that was continuously monitored by the Polly$^{XT}$ system. The lidar measurements confirmed the high variability of the aerosol and its distribution. On some days during the intense measurement period from 1 April to 1 May 2017 the main aerosol load was found in the boundary layer. Those days were connected with low values of the volume linear depolarization ratio. On other days high depolarizing

aerosol was transported over our measurement site. The aerosol layers reached higher altitudes during those events. Signatures of aerosol structures were found up to 9 km altitude. Clouds were frequently embedded within or on top of those aerosol layers.



**Figure 1: Polly$^{XT}$ lidar range corrected signal at 1064 nm (upper plot) and the volume linear depolarization ratio at 532 nm (lower plot) over Limassol, Cyprus from 1 April to 30 April 2017. Layers containing dust and dust mixtures can be identified by the greenish to reddish colours.**

During our measurement period we were able to observe two events of Arabian dust (5 April 2017 and 27-29 April 2017) and one event with a major Saharan dust transport towards our measurement site starting on 20 April 2017 and lasting until 22 April 2017. Similar to the Arabian dust events, the Saharan dust event is characterized by low Ångström exponents indicating almost wavelength independent AOD between at 340 nm and 1020 nm, and with a large contribution of the coarse mode particles to the overall AOD at 500 nm. The maximum AOD during the Saharan dust event was reached on 21 April 2017 with almost wavelength independent values around 0.5 (Ångström exponents < 0.5), during the Arabian dust event at the end of the





measurement period AOD as high as 0.7 was observed. Besides those mineral dust events we were able to characterize two almost pure cases of anthropogenic pollution on 9 April 2017 and on 25 April 2017. While the AOD on 9 April was moderate
with values between 0.1 and 0.25 at 1020 nm and 340 nm, respectively, the AOD during the second event was higher with values of up to ~0.7 at 340 nm, up to ~0.5 at 500 nm, and ~0.15 at 1020 nm. In contrast to the dust cases, the fine mode fraction contributed most to the AOD at 500 nm, while the contribution of the coarse mode particles was almost neglectable with AOD < 0.05. This dominance of the fine mode particles is also reflected in the Ångström exponent which was as high as 1.5. During the other days of the campaign we observed a mixture of different aerosol types, mainly of variable amount of dust and
pollution. Those days were characterized by quite large wavelength dependence of the AOD measurements and large values of the Ångström exponent.

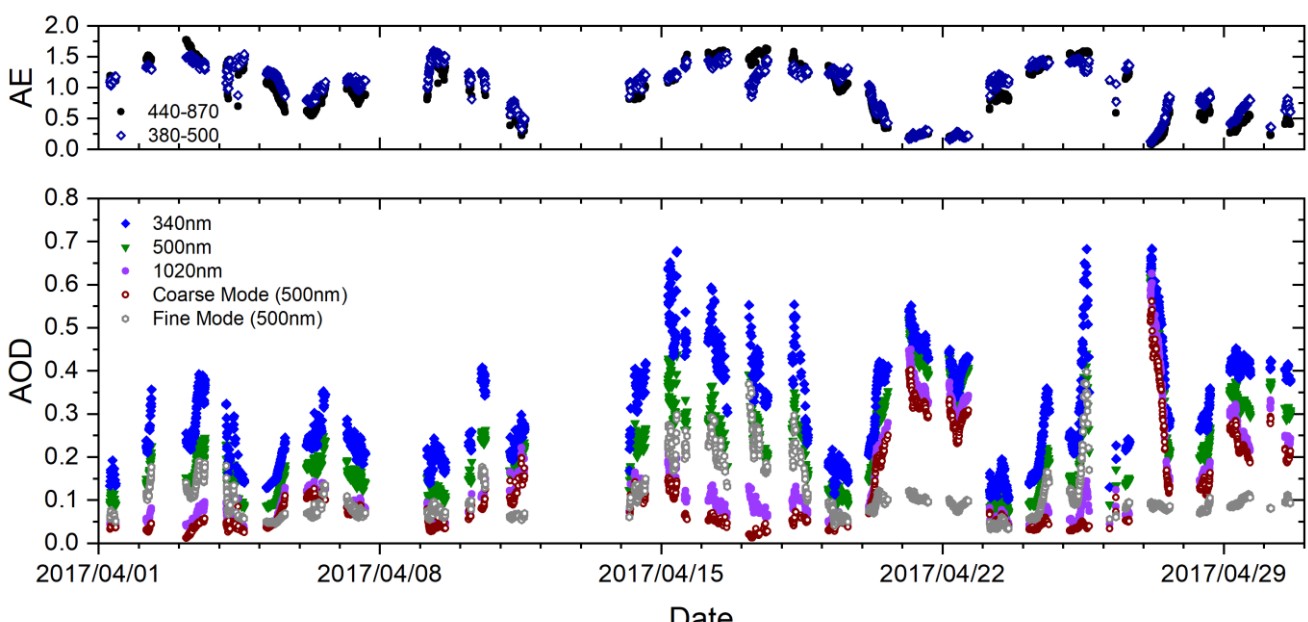

**Figure 2: AERONET sun-photometer measurements and analysis during the A-LIFE field experiment at Limassol, Cyprus showing**
**the aerosol optical depth (AOD) at 340 nm (blue), 500 nm (green) and 1020 nm (red) together with the retrieved coarse mode AOD (brown) and fine mode AOD (grey) at 500 nm (lower panel), the Ångstrom exponent between 440nm and 870 nm (black dots) and between 380 nm and 500 nm (blue diamonds) (upper panel).**






## 3.2 Case studies

In the following we concentrate on three case studies which represent pure Arabian dust (5 April 2017), pollution aerosol (9 April 2017) and Saharan dust (21 April 2017). The main focus of this investigation are the optical properties of pure aerosol types over Cyprus based on lidar measurements. Those analyses are valuable for advanced aerosol typing and to determine the

contribution of different aerosol types to aerosol mixtures. Figure 3 shows the calculated HYSPLIT backward trajectories for the selected case studies.

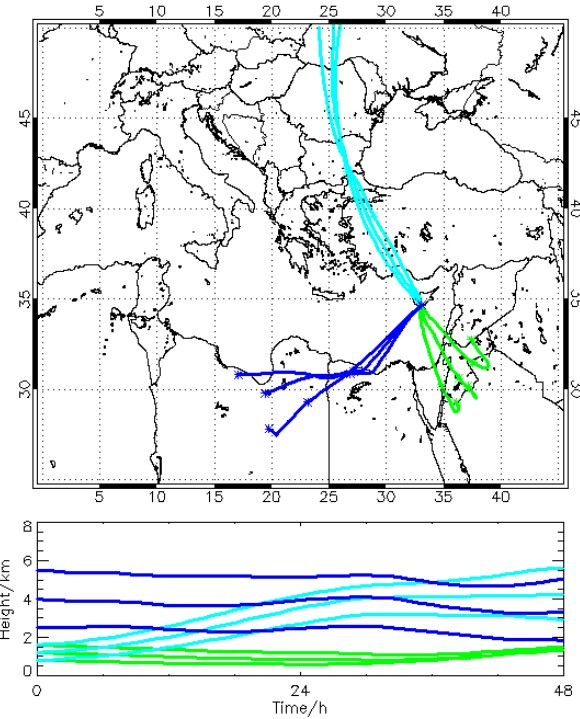

**Figure 3: 48h backward trajectories calculated with the Hybrid Single Particle Lagrangian Integration Trajectory (HYSPLIT) model (Draxler and Rolph, 2012) and Reanalysis meteorological data for the observed aerosol layers on 5 April 2017, 18 UTC (green, Arabian dust), 9 April 2017, 21 UTC (light blue, pollution) and 21 April, 22 UTC (dark blue, Saharan dust). The stars along the trajectories indicate 24 h time steps.**

### 3.2.1 Arabian dust – 5 April 2017 (17:00 – 19:00 UTC)

Already at the beginning of the campaign a dust event with dust aerosols from the Middle East could be observed. Backward trajectories together (Figure 3) with satellite images (not shown) helped to identify the source region of the air masses, which

were advected from southerly directions. The AOD during this event reached values of 0.2 at 500 nm and the situation was characterized by low Ångström exponent of about 0.5. Lidar measurement (Figure 4) show that the main aerosol load was concentrated within the lowest 2 km. The particle linear depolarization ratio (PLDR) at 355 nm and 532 nm show large values





of 0.24 ± 0.02 and 0.27± 0.01, respectively, at a height range between about 0.5 km and 2.0 km. Those values are clear indications of a large dust contribution within the observed aerosol layer (Tesche et al., 2009a; Freudenthaler et al., 2009; Groß

et al., 2011b). The corresponding lidar ratio within this layer shows a wavelength independent value of 40 sr ± 6 sr for 355 nm and 532 nm. Those values are significantly lower than the values found for Saharan dust but agree well with the measurements of a significantly lower lidar ratio of Arabian dust compared to Saharan dust (Mamouri et al., 2013; Nisantzi et al., 2015, Filioglou et al., 2020). The extinction coefficient within the Arabian dust layer shows moderate values of about 0.1 km$^{-1}$ at 355 nm and of about 0.07 km$^{-1}$ at 532 nm. Above the dust layer, the extinction coefficient strongly decreases. In the subjacent

boundary layer, the PLDR values drop to about 0.1 to 0.15 at 355 nm and 532 nm, respectively, indicating that the dust was mixed with a different aerosol type. The corresponding lidar ratio increases to wavelength independent values of about 50 sr to 60 sr, again indicating a change in the aerosol type and mixing state.

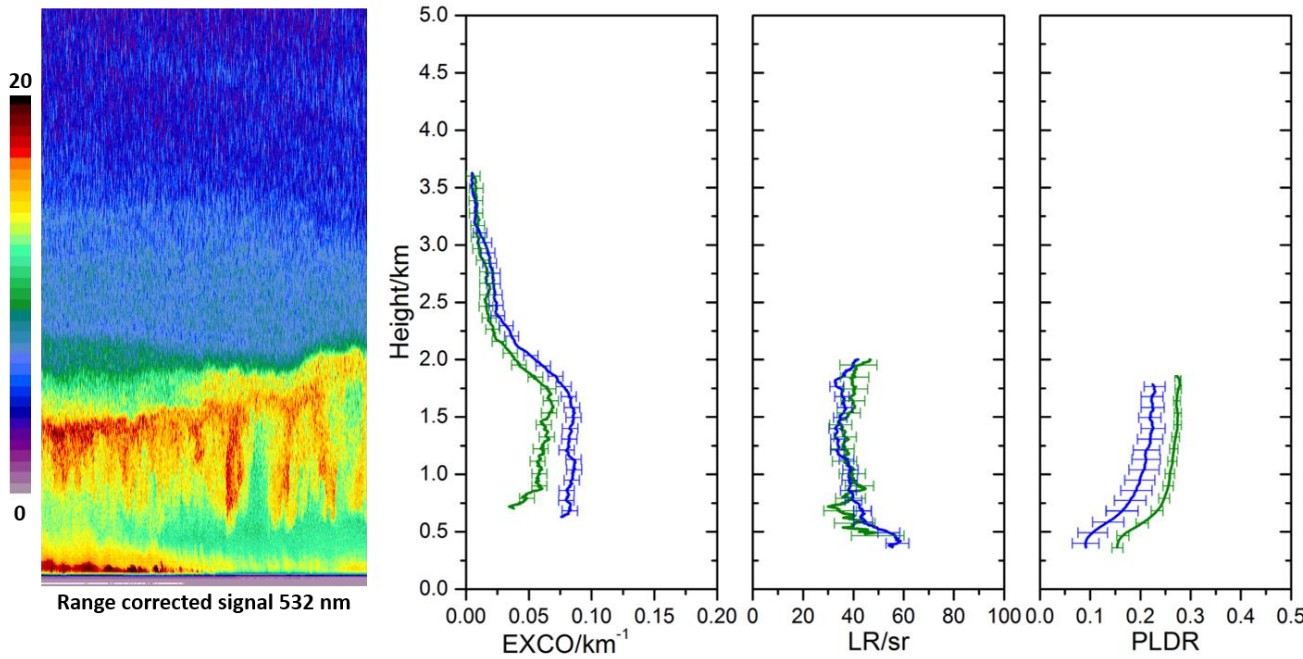

**Figure 4: POLIS lidar measurement showing the range-corrected signal (left panel) in arbitrary units from 17-19 UTC on 5 April**
**2017; the intensity increases from blue over green, orange to red. And profiles of the extinction coefficient (second left panel), the lidar ratio (second right panel) and the particle linear depolarization ratio (right panel). Blue lines indicate measurements at 355 nm, and green lines correspond to measurement at 532 nm. The signals were averaged between 17 UTC and 19 UTC. The error bars show the systematic uncertainty.**





### 3.2.2 Pollution – 9 April 2017 (20:16 – 22:16 UTC)

On 9 April 2017 air masses were advected from north-western directions (Figure 3) towards our measurement site. The situation was characterized by an AOD of about 0.15 at 500 nm together with a large Ångström exponent of about 1.5. These values are clear indications for predominant fine mode aerosols. Lidar measurements between 20:16 UTC and 22:16 UTC are analysed to characterize the optical properties of this aerosol event. The main aerosol load was located within the lowermost 2.0 km (Figure 5). Within a height range of about 0.75 km to 2 km the retrieved extinction coefficient shows a significant wavelength dependence with values >0.1 km$^{-1}$ at 355 nm and values around 0.07 km$^{-1}$ at 532 nm. The values of the retrieved lidar ratio are about 69 sr ± 15 sr wavelength independent between 355 nm and 532 nm. The corresponding PLDR is low with values of about 0.03 ± 0.02 at 355 nm and of 0.04 ± 0.02 at 532 nm. Those values have been reported before for smoke / anthropogenic pollution aerosols (e.g. Groß et al., 2013b; Baars et al., 2021).

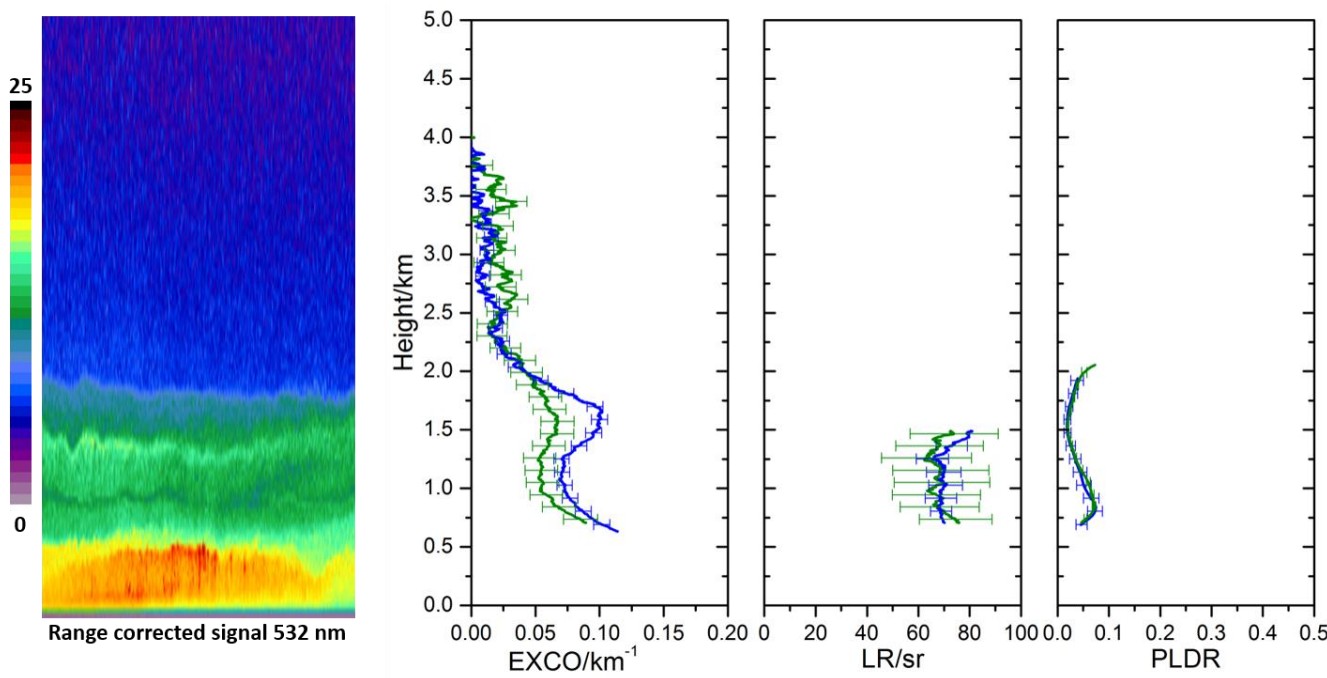

**Figure 5: Same as Figure 4 but for 9 April 2017; 20:16 – 22:16 UTC.**



### 3.2.3 Saharan dust – 21 April 2017 (21:00 – 23:59 UTC)

Between 20 April and 22 April 2017, the aerosol situation over the measurement site was dominated by Saharan dust.
Backward trajectories (Figure 3) indicated the western Saharan regions as main source regions for those aerosol masses. The situation was characterized by an AOD of about 0.5 at 500 nm and low Ångström exponent of about 0.2. The lidar measurements indicate that the top of the aerosol layer reached heights of >6 km (Figure 6). The extinction coefficient within the layer is wavelength independent between 355 nm and 532 nm with maximum values around 0.1 km$^{-1}$ between about 4 km and 6 km height; from 1 km to 3.5 km it is about 0.05 km$^{-1}$. The retrieved lidar ratio and particle linear depolarization are quite
constant with height showing wavelength independent mean values of 59 sr ± 6 sr at 355 nm and of 58 sr ± 8 sr at 532 nm for the lidar ratio and of 0.28 ± 0.03 at 355 nm and 0.29 ± 0.02 at 532 nm for the PLDR. Those values have been reported recently as typical values of Saharan dust after a transport of several days (e.g Groß et al., 2015a; Haarig et al.,2017).

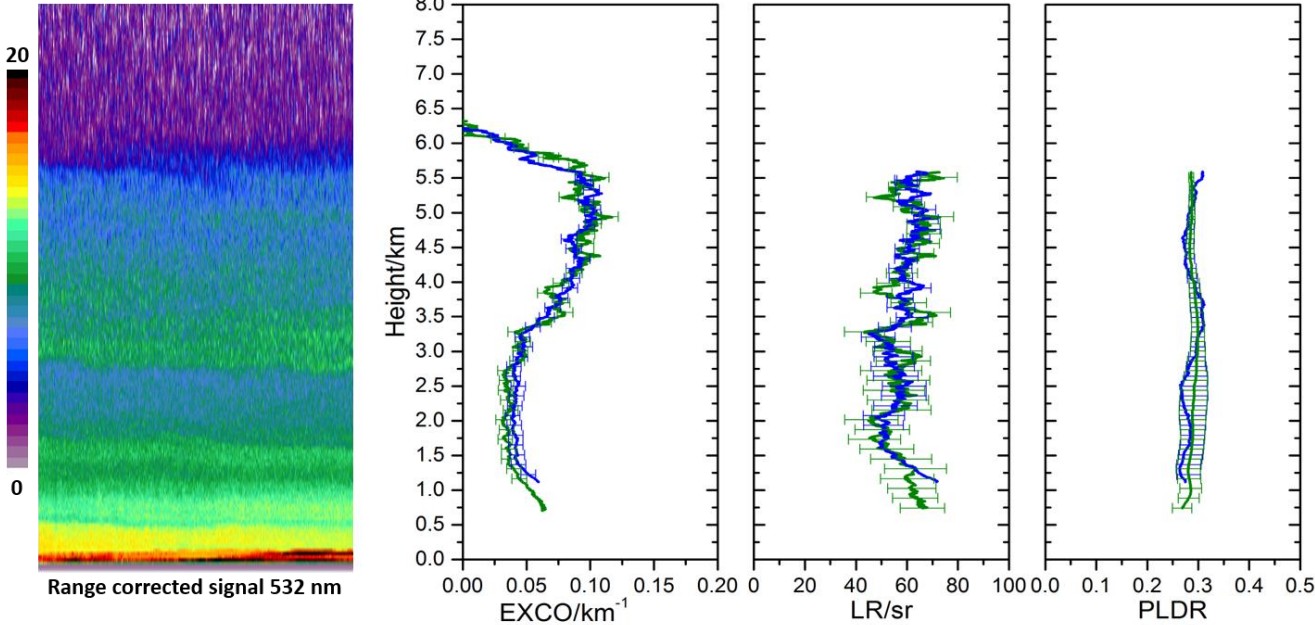

**Figure 6: Same as Figure 4 but for 21 April 2017; 21:00 – 23:59 UTC.**


### 3.4 General findings

As mentioned before, the aerosol situation during the A-LIFE field experiment was highly variable. This was also considered by the flight planning. Measurement flights were performed under different aerosol situations. To better characterize the optical properties and the general aerosol situation with respect to dominating aerosol type during the measurements we analyse the
PLDR and the extinction coefficient for the corresponding heights of the Falcon overpasses over our measurement site (see



the table in the Appendix). As most of the flights were performed during day-time, we were not able to retrieve the lidar ratio. Figure 7 shows the retrieved PLDR and the extinction coefficient for the Falcon overpasses along with the flight altitude. As the signal to noise ratio was too large to retrieve the PLDR for overpasses at flight altitudes >7 km, we restrict our evaluation to the extinction coefficient in those cases. The flight altitudes of most of the overpasses over our measurement site was 1.57

km and about 9.0 km. At the highest flight levels (around 9 km) the extinction coefficients are quite low with values of 0.001 $km^{-1}$ to 0.003 $km^{-1}$ wavelength independent for 355 nm and 532 nm. Lidar and in situ measurements of the extinction coefficient in this height range agree within the estimated measurement/retrieval uncertainty. In the lowermost layer the values range between 0.02 $km^{-1}$ and 0.15 $km^{-1}$. The last value was measured during a strong Arabian dust event at the end of the campaign. The lidar ratio in the corresponding height levels helps to distinguish between different dominating aerosol types.

For overpass 7, 10, 24, 26 and 28 (see Appendix T.1) low PLDR values between 0.03 and 0.13 are found for 355 nm and 532 nm (wavelength independent); for the other days or higher levels, mean values of the PLDR between 0.2 and 0.32 are found at both wavelengths. These large values are a clear indication that the layer has a strong contribution of mineral dust particles or that mineral dust was even the only aerosol type in this layer. The PLDR at 532 nm within the layer is used to derive the contribution of dust and non-dust (assuming anthropogenic pollution) of the extinction coefficient at 532 nm. During the strong

Saharan dust event from 20-22 April and during the strong Arabian dust event at 27-29 April, those analysis show that dust is by far dominating the extinction coefficient of the layer and the contribution of anthropogenic pollution is only minor with values of max. 0.01 $km^{-1}$. On days with low mean PLDR values (~0.05) at flight altitude, anthropogenic pollution is dominating the extinction coefficient at 532 nm within this layer while dust has only a minor contribution.



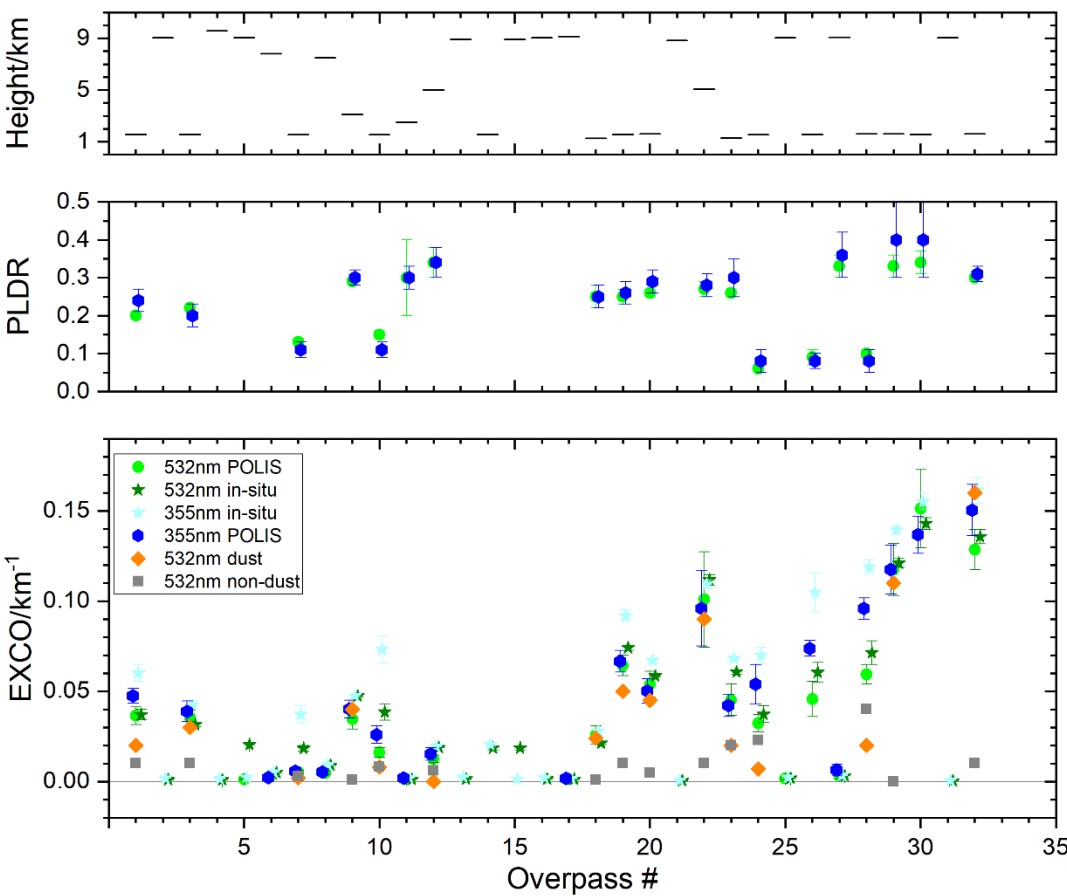

**Figure 7: Falcon flight altitude (upper panel) used for the analyses of the PLDR (middle panel) at 355 nm (blue) and 532 nm (green) and the extinction coefficient (EXCO) from POLIS at 355 nm (blue) and 532 nm (green), for in-situ measurements at 355 nm (cyan) and 532 nm (olive) and for the dust (orange) and non-dust (grey) extinction coefficient at 532 nm. The error bars give the systematic uncertainties.**

## 3.5 Aerosol typing

One main goal of the A-LIFE field experiment was the characterization of the aerosol situation during the spring season during which aerosol mixtures of natural (mineral dust) and anthropogenic pollution from different origins are frequently observed. The campaign was designed as a closure study combining different measurement techniques, including ground-based and airborne in-situ and remote-sensing measurements. To investigate how well we can classify the different aerosol types and mixtures with lidar and sun-photometer, how this is pictured by transport simulations, how well it agrees with in-situ measurements, we compare the different techniques with one another.




### 3.5.1 AERONET

For the AERONET based aerosol typing we use the scatter plot of Ångström exponent vs. AOD at 500 nm as proposed by
Toledano et al. (2009, 2011). Values of the Ångström exponent (AE) larger than ~1.2 serve as indication for smoke / anthropogenic pollution. The AOD for those pollution cases can vary between about 0.1 to quite large values. Ångström exponents of about <0.5 serve as indication for dust (Toledano et al., 2009, 2011, 2019) or marine aerosols. A threshold of AOD=0.15 is used to separate dust and marine aerosols. Ångström exponents between about 0.5 and 1.2 serve as indication for mixtures of dust with other aerosol types (e.g. smoke or pollution). Figure 8 shows the scatter plot of the sun-photometer
measurements during the A-LIFE field experiment. Marine aerosol scenes were not observed as low Ångström exponents with corresponding low AOD values are missing. Low Ångström exponents during A-LIFE came along with large AOD. This is a typical signature of mineral dust events. Those events are classified as dust. Large values of the Ångström exponent with corresponding AOD of about 0.2 to 0.4 are also frequently found during A-LIFE, clearly indicating a dominance of anthropogenic pollution or biomass burning aerosols during those days. All other days show intermediate values which
correspond to aerosol mixtures with varying contribution of dust and pollution/smoke. This AOD-AE plot cannot be used to distinguish between Saharan dust and Arabian dust. Dedicated analysis to this difference is given by Mateos et al. (2024) using sun-photometer inversion products. Along with the AOD-AE space, Figure 8 also shows the measurement in the AOD-Coarse Mode AOD space. This plot shows different arms of the distribution. Low Coarse Mode AOD along with low to moderate AOD are found for pollution, which was also confirmed by looking at the AOD-AE measurements. Measurements of medium
(0.2) to large Coarse Mode AOD together with medium to large AOD values are indications for mineral dust, and the rest of the values are found for mixtures of dust and pollution. If the dominating aerosol type is dust in those mixtures, the Coarse Mode AOD is slightly higher than for the mixtures with a dominance of pollution. The corresponding classification for the Falcon overpasses is listed in Table 2.

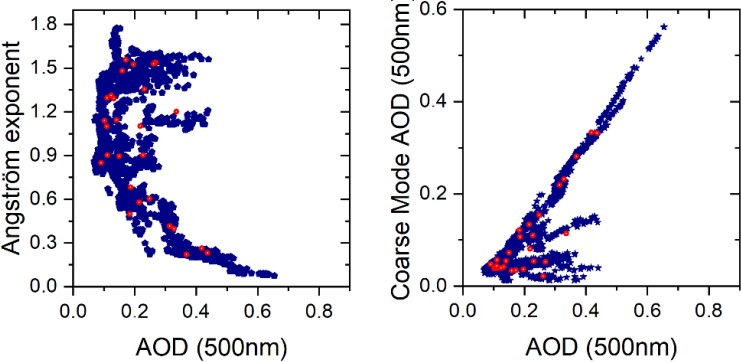

**Figure8: AERONET sun-photometer measurements and analysis during the A-LIFE field experiment at Limassol, Cyprus showing the aerosol optical depth (AOD) at 500 nm (green) vs the Ångström exponent (left), and vs. the Coarse Mode AOD at 500 nm (right). Blue stars indicate the individual measurements, red dots give the daily mean value.**



### 3.5.2 Lidar

The mean values of the lidar ratio and the PLDR for the different aerosol layers and different aerosol types measured with the

lidar systems POLIS and Polly[XT] during night time are shown in Table 1 and Figure 9. The values show the large variability of the aerosol composition over the measurement site at Limassol, Cyprus. For the aerosol typing based on lidar measurements we use a method proposed by Burton et al. (2012) and Groß et al. (2013a, 2015b). This method depends on the fact that the lidar ratio and the PLDR are quite different for different aerosol types. Up to now, those schemes did not include the Arabian dust. Thus, our measurements will expand these classification schemes by another aerosol type of interest in large parts of the

globe. Figure 9 shows the PLDR vs. lidar ratio at 355 nm and 532 nm. In the background of both plots the measurements from former campaigns, that are already included in the typing schemes, (Groß et al., 2015a, b) are shown. In the foreground (large symbols) the measurements during this campaign are shown. The measurements during the days dominated by anthropogenic pollution and Saharan dust clearly fit in the former classification scheme. Thus, the aerosol type within these layers can be clearly classified. Arabian dust has lower values of the lidar ratio compared to Saharan dust, both at 355 nm and 532 nm. This

was also confirmed by Filioglou et al. (2020). They found values of the lidar ratio of about 42-45 sr in lidar measurements at 355 nm and 532 nm and corresponding PLDR values of about 0.25 at 355 nm and of about 0.31 at 532 nm. For Asian dust similarly, low values of the lidar ratio were found by Hofer et al. (2017, 2020), which reported values of 39-45 sr for 355 nm and 532 nm with PLDR of about 0.24 at 355 nm and of about 0.33 at 532 nm, while Hu et al. (2020) found larger values of the PLDR for measurements near the Taklimakan desert. They interpreted these large values as an indication for fresh dust

close to the source regions with a large amount of coarse and giant particles. PLDR of less than ~0.07 at 355 nm and 532 nm together with high lidar ratios of about 60-75 sr at both wavelengths indicated pollution aerosol layers. In addition, the lidar measurements indicated different layers with aerosol mixtures. Those layers are indicated by intermediate values of the PLDR and the lidar ratio. Pure marine aerosol layers could not be identified from the lidar measurements during A-LIFE; they rather indicate mixtures with dust and/or pollution.






**Table 1: Mean values of the lidar ratio (LR) and particle linear depolarization ratio (PLDR) at 355 nm and 532 nm including the mean systematic errors (±) for different aerosol days and height ranges. For this evaluation night-time POLIS measurements are used.**

| Date | Height (km) | LR355 (sr) | | LR532 (sr) | | PLDR355 | | PLDR532 | |
|---|---|---|---|---|---|---|---|---|---|
| | | POLIS | POLLY | POLIS | POLLY | POLIS | POLLY | POLIS | POLLY |
| 5 April | 0.7-2.0 | 40 ± 6 | 39 ± 6 | 40 ± 6 | 32 ± 5 | 0.24 ± 0.02 | 0.24 ± 0.02 | 0.27 ± 0.01 | 0.27 ± 0.03 |
| 6 April | 1.5-3.0 | 33 ± 11 | 45 ± 7 | 44 ± 11 | 41 ± 7 | 0.10 ± 0.02 | 0.09 ± 0.02 | 0.14 ± 0.02 | 0.12 ± 0.02 |
| 9 April | 0.7-1.5 | 69 ± 15 | | 69 ± 15 | | 0.03 ± 0.02 | 0.07 ± 0.03 | 0.04 ± 0.02 | 0.04 ± 0.02 |
| 11 April | 1.0-1.4 | | 68 ± 13 | | 66 ± 32 | | 0.12 ± 0.02 | | 0.04 ± 0.02 |
| 14 April | 0.9-1.2 | | 35 ± 6 | | 22 ± 9 | | 0.04 ± 0.02 | | 0.02 ± 0.02 |
| 20 April | 3.0-4.5 | 61 ± 7 | 49 ± 8 | 62 ± 7 | 45 ± 9 | 0.28 ± 0.02 | 0.22 ± 0.02 | 0.29 ± 0.01 | 0.28 ± 0.03 |
| 21 April | 3.0-5.5 | 59 ± 6 | 50 ± 8 | 58 ± 8 | 50 ± 8 | 0.28 ± 0.03 | 0.23 ± 0.03 | 0.29 ± 0.03 | 0.28 ± 0.02 |
| 22 April | 1.5-3.0 | 48 ± 10 | | 45 ± 4 | | 0.29 ± 0.03 | | 0.29 ± 0.01 | |
| 25 April | 0.7-2.0 | 60 ± 10 | 60 ± 10 | 60 ± 10 | 61 ± 11 | 0.03 ± 0.03 | 0.06 ± 0.02 | 0.04 ± 0.03 | 0.06 ± 0.02 |
| 26 April | 2.0-2.8 | | 40 ± 5 | | 40 ± 5 | 0.30 ± 0.03 | 0.28 ± 0.02 | 0.31 ± 0.02 | 0.32 ± 0.03 |
| 27 April | 1.5-3.0 | 42 ± 3 | 39 ± 7 | 45 ± 3 | 34 ± 5 | 0.30 ± 0.03 | 0.27 ± 0.03 | 0.30 ± 0.02 | 0.32 ± 0.03 |
| 29 April | 0.9-2.0 | | | | | 0.26 ± 0.03 | | 0.28 ± 0.02 | |






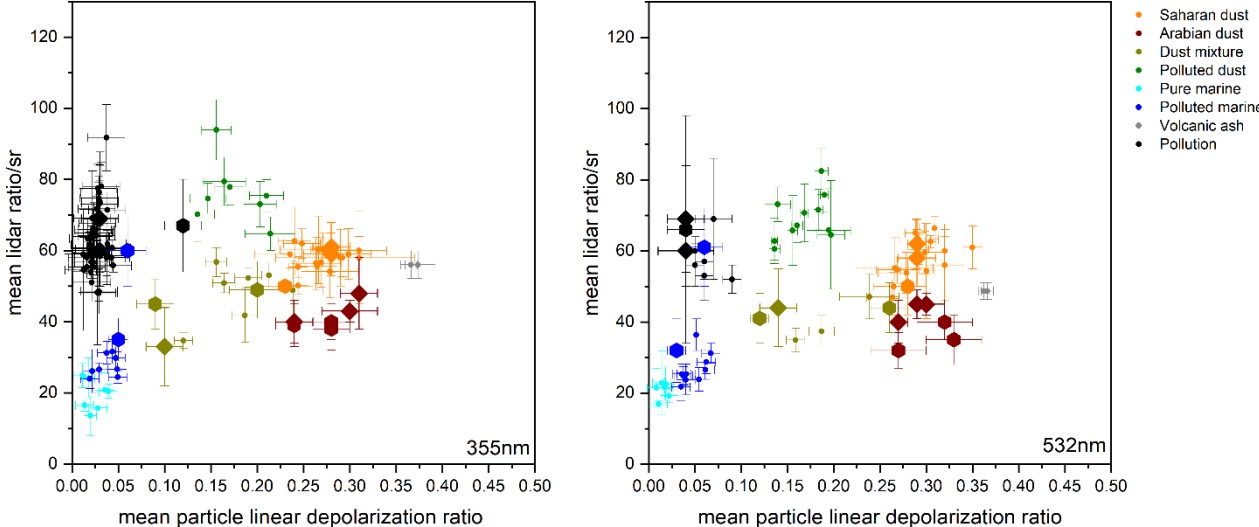

**Figure 9: Aerosol classification at 355 nm (left) and 532 nm (right) based on the particle linear depolarization ratio and the lidar ratio. Measurements given by small symbols show findings from former studies (Groß et al., 2015a, b), measurements given by large symbols show measurements during A-LIFE; diamonds show POLIS measurements, hexagons show PollyᵡT measurements. The figure is adopted from Groß et al., 2015a. Light green and blue data points in the background indicate fresh biomass burning aerosol and marine aerosol, respectively.**

## 4 Discussion

### 4.1 Comparison of aerosol typing methods

Aerosol type classification is one important point in determining the radiative properties of the layers as well as to estimate possible interactions, e.g. with clouds. Different aerosol type classification schemes are based on different measured properties, i.e. microphysical properties vs. optical properties (Section 2 and Section 3.5). Thus, the different methods might give slightly different results, depending also on the number and sub-types involved. To ensure that the different methods included in A-LIFE (in-situ, lidar, sun-photometer, transport simulations) give the same results, we intercompare the output of those methods for selected overpasses. As we do not have the lidar ratio for the aerosol type classification during day-time, we classify the aerosol type at flight level by a combination of different pieces of information. We analyse the next Raman measurements at night-time together with an evaluation of the stability of the aerosol situation. This allows to use the classification at night-time as a first proxy for the day-time measurements. This first guess is then confirmed by the continuity of the PLDR and verified by the evaluation of air mass source regions. The lidar classification is not considered when we do not have collocated measurements in the flight altitude or cannot do an aerosol classification due to temporal variability or low signal to noise ratio.



Table 2 gives an overview of the aerosol type classification of all used classification methods for the selected Falcon overpasses over the Limassol site. The different methods agree well in the classification of the dominating aerosol type, but it is also

obvious that small differences occur. AERONET sun-photometer measurements can only provide a classification for the whole atmospheric column due to the measurement setup. Thus, at days with different aerosol layers consisting of different aerosol types, the combination will be reflected in the classification. This is the case for the complex situation on 5 April, 6 April and 11 April, for example. If only one main aerosol type is present, the classification is well comparable with the height resolved classification from the lidar and the in-situ measurements. Comparing the lidar and in-situ classifications, we also see a good

agreement in general. However, minor contributions (e.g. from pollution) cannot be characterized with the lidar when the optical properties of the layer are strongly dominated by dust aerosols. This is shown e.g. for the strong Arabian dust outbreak on 27 April and 29 April. As another source of aerosol type characterization, we use atmospheric transport calculations with the FLEXPART model in this study. This method considers a larger number of different aerosol types than derived from AERONET, lidar and in-situ classification (including source allocation). We see that the FLEXPART-based classification in

principle fits well the in-situ and lidar classifications



**Table 2: Date, Time, and Height of Falcon overpasses over the Limassol measurement site together with resulting aerosol classification from in-situ, AERONET and lidar measurements and atmospheric transport simulations with FLEXPART. AD stands for Arabian Dust, SD for Saharan Dust, OM for Organic Matter, SS for Sea Salt, SO4 for Sulphate Aerosols, and CM for Coarse Mode.**

| Date | Time/UTC | Heigh/km | in-situ | AERONET | Lidar | FLEXPART |
|---|---|---|---|---|---|---|
| 5 April | 8:50 | 1.57 | Polluted AD | Dust and pollution | AD | AD, OM, SS |
|  | 11:13 | 9.03 | Polluted mixture (low CM) |  | - |  |
| 6 April | 4:30 | 1.57 | Moderately polluted SD | Dust and pollution | Dust mixture (marine) | SD, OM, SO4, SS |
|  | 7:30 | 9.57 | Polluted mixture (low CM) |  | - | - |
| 11 April | 5:07 | 1.54 | Polluted mixture (enhanced CM) | Mixed dust | Mixed pollution | OM |
|  | 6:15 | 7.48 | Pure SD |  | - | - |
|  | 6:58 | 3.11 | Pure SD |  | Dust | SD |
|  | 8:24 | 9.04 | Moderately polluted SD |  | - | - |
|  | 8:33 | 7.8 | Moderately polluted SD |  | - | - |
| 14 April | 04:13 | 1.55 | Moderately polluted AD | Polluted mixture | Mixed pollution | AD, SD, SS, OM |
| 14 April | 11:37 | 8.91 | Polluted mixture (low CM) |  | - | - |
| 20 April | 18:21 | 1.25 | Polluted SD | Dust | SD | - |
| 21 April | 11:52 | 1.57 | Moderately polluted SD | Dust | SD | SD, OM, SS |
|  | 15:27 | 9.44 | Moderately polluted mixture (low CM) |  | - |  |
| 22 April | 06:10 | 1.59 | Moderately polluted SD | Dust | SD | SD, OM |
|  | 06:35 | 8.81 | Moderately polluted SD (low CM) |  | - |  |
|  | 07:27 | 5.06 | Pure SD |  | SD | SD |
| 25 April | 8:07 | 1.54 | Polluted mixture (enhanced CM) | Pollution | Mixed pollution | OM, SO4, SS (dust) |
|  | 9:50 | 9.03 | Moderately polluted SD |  | - | SD |
| 27 April | 7:17 | 1.57 | Moderately polluted AD | Dust | AD | AD, OM, |
|  | 8:47 | 9.05 | Moderately polluted mixture (low CM) |  | - |  |
|  | 9:57 | 1.58 | Moderately polluted AD |  | AD | AD, OM |
| 29 April | 7:09 | 1.58 | Moderately polluted AD | Mixed dust | AD | AD, OM |





## 4.2 Dust and non-dust fraction

From the aerosol typing comparison above we find that the different methods agree quite well for aerosol typing. However, to better characterize the aerosol situation and thus investigate the impact of the different aerosols, not only the aerosol type but

also the fractional contribution of a specific aerosol type to the optical properties or the volume concentration as well as its mass concentration is of importance. To intercompare the different methods with respect to the given dust mass concentration, we first calculate the dust fraction of the backscatter coefficient and of the volume concentration and the dust mass concentration for the different overpasses (Figure 10), and compare the latter to the estimated dust mass concentration from the FLEXPART simulation as well as the total mass concentration.

The contribution of the dust fraction to the backscatter coefficient at 532 nm is similar to the dust contribution to the extinction coefficient (Figure 7). During the major dust events the dust fraction of the layer mean backscatter coefficient varies between about 0.8 at the first events of Arabian dust at the beginning of the campaign to 1.0 at the strong Arabian dust event at the end of the campaign. For the Saharan dust event the dust fraction of the layer mean backscatter coefficient was about 0.9. And even during the days with a dominance of anthropogenic pollution, the dust fraction of the backscatter coefficient is still about

0.3, except during the pure pollution event where we find dust fractions of the layer mean backscatter coefficient <0.1. The derived volume fraction of dust aerosols follows the dust fraction of the optical properties. Thus, in general, one can conclude, that the dominance in the optical properties is a result of the dominance in volume.

Derived mean dust mass concentrations from the lidar measurements at 532 nm at flight altitude reflect the large variability during the measurement period. Large values of dust mass concentration at flight altitude of around $300\mu g/m^3$ are found during

the strong Arabian dust event at the end of the campaign. The backscatter and extinction coefficients at flight altitude at those days are also quite large, which perfectly fits to the large values of the dust mass concentrations. During all other dust events we find dust mass concentrations between about $50\mu g/m^3$ during the first Arabian dust event and about $170\mu g/m^3$ during the major Saharan dust event. For measurements with dust and non-dust mixtures we find a dust mass of $<35\mu g/m^3$ or even none during the pollution events. Comparing the lidar derived dust mass concentrations with the calculated dust mass concentrations

from FLEXPART, we find a good agreement for the low and moderate dust cases; e.g. the Arabian dust event in the beginning of the campaign or for the moderately Saharan dust event around the 11 April 2017. In contrast, FLEXPART was not able to reproduce the dust mass concentrations for the strong dust events like the Saharan dust event around the 21 April 2017, when FLEXPART estimated only about 50 $\mu g/m^3$ compared to about 170 $\mu g/m^3$ derived from the lidar measurements. The disagreement is even worse for the strong Arabian dust event at the end of the campaign. The dust mass concentration derived

from the lidar measurements shows values as high as 285 $\mu g/m^3$, while FLEXPART estimates only about 30 $\mu g/m^3$. Due to the large discrepancies between the lidar derived dust mass concentration and the FLEXPART estimated dust mass concentration, we use the in-situ measured total mass concentration. Comparing now the three methods one can see, that the total mass concentration has in general a better agreement with the lidar derived dust mass concentration. That confirms the



large mass concentration during the strong Saharan dust event around the 21 April and the strong Arabian dust event at the
end of the campaign that are derived from the lidar measurements. The total mass concentration even exceeds the lidar derived
dust mass concentration for most of those days. Differences are caused by the fact that both quantities are not completely the
same and, e.g. the averaging time of the lidar and the in-situ measurements are different and thus the sampled volume.

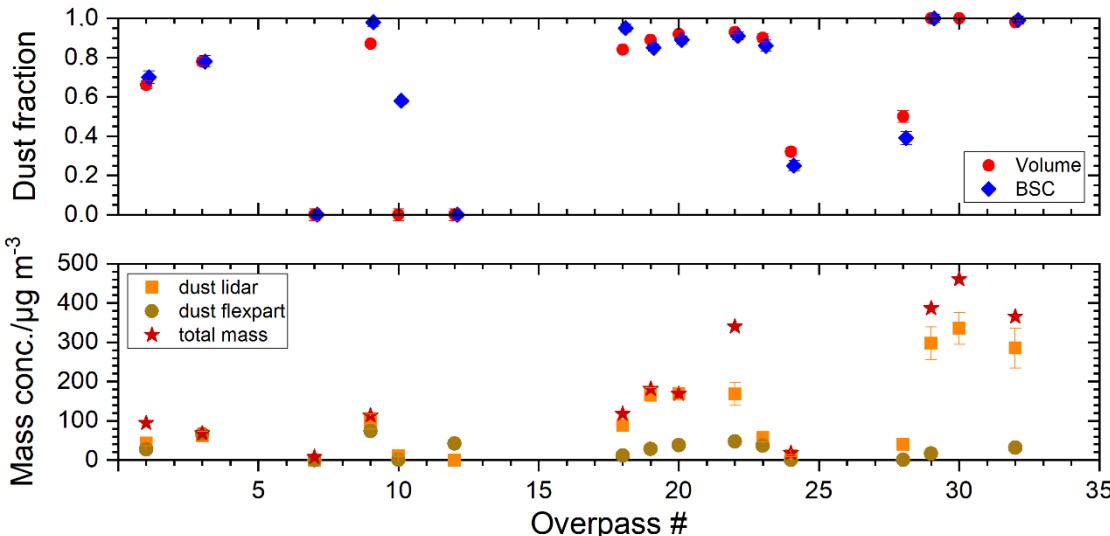

**Figure 10: Retrieved volume (red) and backscatter coefficient (blue) dust fraction derived from POLIS lidar measurements (upper
panel), and dust mass concentration (lower panel) derived from POLIS lidar measurements (orange), FLEXPART (light brown),
and in-situ total mass concentration (red stars) for the Falcon overpasses.**

## 5 Summary and Conclusion

In this study we investigated the optical properties of complex mineral dust and other absorbing aerosol mixtures in the Eastern
Mediterranean. We found significantly lower values of the lidar ratio with means of 41 sr ± 5 sr at 355 nm and 39 sr ± 5 sr at
532 nm for Arabian dust transported to our measurement site, compared to the lidar ratio found for Saharan dust. These findings
are in good agreement with previous lidar studies (Mamouri et al., 2013; Nisantzi et al., 2015). The PLDR of Arabian dust is
similar to what was found for Saharan mineral dust close to the source region (e.g. Tesche et al., 2009b; Freudenthaler et al.,
2009; Groß et al., 2011b) with mean values of 0.27 ± 0.02 at 355 nm and 0.27 ± 0.02 at 532nm. The Saharan and Arabian
mineral dust layers could be clearly distinguished from other aerosol layers with pollution by means of their PLDR of 0.27 ±
0.02 at 355 nm and 0.28 ± 0.02 at 532 nm and their lidar ratio of 55 sr ± 8 sr at 355 nm and 53 sr ± 7 sr at 532 nm. The lidar
derived optical properties found for Saharan and Arabian dust are in good agreement with the values found for transported



Saharan dust (e.g. Groß et al., 2015a; Haarig et al., 2017). For pollution aerosol we found mean values of the PLDR and lidar ratio of $0.05 \pm 0.02$ and $65$ sr $\pm 12$ sr at 355 nm and $0.05 \pm 0.02$ and $60$ sr $\pm 16$ sr at 532nm. Those values of the PLDR and the

lidar ratio for pollution aerosol confirm the values reported by Groß et al. (2015a) and the papers cited therein. We compared the findings of the lidar based classification to results from other aerosol typing methods, i.e. methods based on in-situ measurements, sun-photometer, and FLEXPART transport simulations. The different classification schemes showed a very good agreement, although the sun-photometer based classification can only give a column integrated value.

We frequently found that pollution aerosol was mixed into the dust layers. Nevertheless, the lidar derived extensive optical

properties (i.e. extinction coefficient and backscatter coefficient) were dominated by mineral dust during significant dust events. The derived volume fraction of the dust aerosols partly showed a lower contribution to the total volume compared to its contribution to the optical properties. The derived dust mass concentration varied strongly throughout the measurement period. The highest values of about $170$ µg/m$^3$ and of about $300$ µg/m$^3$, derived from lidar, were found during a major Saharan and Arabian dust event, respectively. While the FLEXPAT derived dust mass concentration agreed quite well with the lidar

derived dust mass concentration for low and moderate dust load, FLEXPART could not reflect the high dust concentrations during strong mineral dust events; although it could predict the dust transport in general. Models generally assume that dust aerosols are spherical or spheroidal, this leads to the assumption of more gravitational settling and thus helps explain the underestimation of coarse dust transport (Huang et al., 2020). In order to improve the confidence in the high dust mass concentration derived from lidar measurements during these events, we compared them to the in-situ derived total mass

concentration. During our measurement period we found a general good agreement of the total mass concentration and the lidar derived dust mass concentration. However, during strong dust events the in-situ derived total mass concentration exceeded the lidar derived dust mass concentration.

**Author contributions**

BW coordinated the A-LIFE project. BW, MD, JG and MT performed the in-situ measurements. BW, MD, JG, MT and MS

analysed the in-situ data. AT and PS performed the FLEXPART simulations. SG, VF, AA performed the lidar measurements. SG, MH and CU analysed the lidar data. CT, DM performed and analysed the sun photometer measurements. RM and AN supported the lidar and sun photometer measurements. SG wrote the manuscript. All authors discussed the data and findings.

**Competing interests**

The contact author has declared that none of the authors has any competing interests



**Acknowledgements**

The A-LIFE field experiment was mainly funded by an ERC Starting Grant (A-LIFE) with support of the Deutsches Zentrum für Luft- und Raumfahrt (DLR). The lidar measurements were funded by TROPOS, DLR and the Ludwig-Maximilians-Universität München, and the sun photometer measurements by the University of Valladolid. We thank the Institute for Flight Experiment for conducting the research flights, and we are grateful to Daniel Sauer (DLR) for helpful comments on the
manuscript. The research has been supported by the European Commission (Project 101137680 – CERTAINTY) and by DLR internal funding within the MABAK projet.

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





780    Appendix - Overpasses

| Date | Start time / UTC | Height / km | Overpass # |
|------|------------------|-------------|------------|
| 170405 | 170405 08:52 | 1,57 | 1 |
| 170405 | 170405 11:13 | 9,03 | 2 |
| 170406 | 170406 04:33 | 1,57 | 3 |
| 170406 | 170406 07:31 | 9,56 | 4 |
| 170411 | 170411 08:24 | 9,04 | 5 |
| 170411 | 170411 08:33 | 7,8 | 6 |
| 170411 | 170411 05:07 | 1,54 | 7 |
| 170411 | 170411 06:15 | 7,48 | 8 |
| 170411 | 170411 06:58 | 3,11 | 9 |
| 170411 | 170411 12:46 | 1,54 | 10 |
| 170411 | 170411 10:01 | 2,48 | 11 |
| 170411 | 170411 11:30 | 4,99 | 12 |
| 170413 | 170413 11:10 | 8,92 | 13 |
| 170414 | 170414 04:13 | 1,55 | 14 |
| 170414 | 170414 11:37 | 8,91 | 15 |
| 170419 | 170419 17:57 | 9,03 | 16 |
| 170420 | 170420 17:38 | 9,13 | 17 |
| 170420 | 170420 18:21 | 1,25 | 18 |
| 170421 | 170421 11:52 | 1,57 | 19 |
| 170422 | 170422 06:10 | 1,59 | 20 |



| 170422 | 170422 06:35 | 8,81 | 21 |
|--------|--------------|------|----|
| 170422 | 170422 07:27 | 5,06 | 22 |
| 170422 | 170422 08:24 | 1,28 | 23 |
| 170425 | 170425 08:07 | 1,54 | 24 |
| 170425 | 170425 09:50 | 9,03 | 25 |
| 170426 | 140726 12:09 | 1,56 | 26 |
| 170426 | 170426 13:07 | 9,06 | 27 |
| 170426 | 170426 14:26 | 1,58 | 28 |
| 170427 | 170427 09:57 | 1,58 | 29 |
| 170427 | 170427 07:17 | 1,57 | 30 |
| 170427 | 170427 08:47 | 9,05 | 31 |
| 170429 | 170429 07:09 | 1,58 | 32 |