# Peer review of "Characterization of aerosol over the Eastern Mediterranean by polarization sensitive Raman lidar measurements during A-LIFE – aerosol type classification and type separation"

_EGUsphere, 2024_

## Author Comment (AC1)

**We thank the reviewers for the careful reading and the suggestions that helped us improve our manuscript. The answers to the reviewers will be given after each comment in bold font.**

**Reply to Referee #2**

**general comments**

The manuscript entitled „Characterization of aerosol over the Eastern Mediterranean by polarization sensitive Raman lidar measurements during A-LIFE – aerosol type classification and type separation" by Groß et al. promises an important contribution in the field of aerosol typing from different measurement techniques. Nevertheless, I feel slightly disappointed after reading the manuscript for two main reasons.

1. It seems like parts of the manuscript have been written by different people. Some sections are well written, others are quite sloppy in usage of language and presentation of figures.

**We carefully revised the manuscript to hopefully achieved to better homogenize the different parts of the manuscript and improve insufficient usage of language and presentation.**

2. The abstract describes the goal of the A-LIFE campaign to "characterize dust in complex mixtures". But a detailed discussion of aerosol mixtures is missing in the paper. Instead the case studies focus -again - on situations with pure aerosol types and the discussion of derived aerosol types and dust fractions is not comprehensive enough.

**We believe that showing this pure aerosol cases is important for the significance of the investigation of mixtures later in this paper. We want to make sure – and show – that optical properties different from the know values for pure cases are really due to mixture and not due to transport or aging. Thus, we prefer to keep the case studies as they are. However, we agree, that the discussion of the derived aerosol types and the dust fraction could be extended and improved. We revised these parts of the manuscript.**

**Some important points for improvement**

**Section 2.2:**

It is unclear how the extinction coefficients from day-time measurements (presented in Fig. 7) were derived. Please, elaborate in more detail. Furthermore, concerning the retrieved uncertainties, it is insufficient to just provide a reference to previous work. Especially since the captions of Figures 4-6 claim that all error bars only show systematic uncertainties, however the references clearly describe the handling of statistical errors as well. Please, add some sentences like "systematic errors of lidar ratio include the uncertainties of backscatter calibration, …, "

**For the day-time measurements the extinction coefficient is retrieved with the Fernald-Klett algorithm, as described in the Methodology section. As described, this algorithm uses the lidar ratio as an input.**

We did not include the statistical errors as those are only available for the Raman measurements and were not included in the references for the depolarization measurements. We would have to come up with a new method of error propagation for the statistical error of the depolarization measurements. Furthermore, the statistical uncertainty strongly depends on the averaging length. However, we included a short description of the what is included in the systematic errors.

**Sections 2.3 – 2.5**:

The reader may get the impression that data of both lidar systems have been analyzed with the same software tools, and/or by the same experts. Please, elaborate.

**We modified the description of the systems to make clear that this is not the case.**

**Section 2.6**:

Please, provide technical details (e.g. thresholds) of the aerosol typing here, instead of in section 3.5.1.

**We moved the technical details from section 3.5.1 to the Methodology section and extended the description of the aerosol typing (including thresholds).**

**Section 2.7**:

Since the publication by Weinzierl et al. is not yet published, the reader needs more details here, not just a reference. Such details should entail: what exactly are the derived aerosol type "mixtures?" What components do those mixtures contain? It is also entirely unclear, what a (moderately) polluted mixture with or without coarse mode might be. Please, provide descriptive examples like "sea salt mixed with smoke". To that point, it is unclear whether the in-situ classification scheme depends on FLEXPART input or not. If it is not independent, why provide the comparison in Table 2?

**We expected the publication by Weinzierl et al. to be published yet. As this is not the case, we removed the reference to this publication. However, a description of the method and also of the complex mixtures of dust and pollution is given by Teri et al., 2022 and 2024. Both publications are cited in our study. A detailed evaluation of mixtures and how that effects the in-situ measurements and optical properties is given in Teri et al., 2024 (doi.org/10.5194/egusphere-2024-701). It is beyond the scope of this study to repeat it here. However, we included some further information of the in-situ measurements for days classified as mixtures from the in-situ measurements and as pure from the lidar measurements in a table in the supplement.**

**Section 2.8**:

Is it possible to provide thresholds for the aerosol typing here?

**We just used the pre-classification. For details on how the typing is actually done we have to refer to the publications describing FLEXPART.**

**Section 2.9**:

If the highly sophisticated FLEXPART simulations were available for the campaign, why is it necessary to work with basic HYSPLIT trajectories, too? Please, elaborate why FLEXPART simulations were not used for analysis of the lidar measurements.

**The FLEXPART simulations were run for the aircraft measurements (flight track and altitude). Thus, they were not available for the full lidar profile. As it was not the intention of this study to include a full investigation of the source region and transport of the aerosols we used HYSPLIT to highlight, that the airmasses for the different case studies were indeed advected from different regions.**

**Section 3.1**:

It is mentioned here that smoke aerosols have been present during the campaign. But section 2.4 does not describe the handling of smoke in the aerosol typing procedures, please elaborate.

**Smoke and pollution cannot be distinguished from the remote sensing instrumentation as they are quite similar in the measured optical properties. That is why we referring to pollution/smoke throughout the text. As that, it is included in Section 2.4 and after a revision of the aerosol typing from AERONET also in section 2.7**

**Figure 1:**

Change minor ticks in a way that each interval corresponds to one day. Indicate Falcon overpass times and altitudes by symbols and times of lidar case studies by vertical lines

**We changed that.**

**Figure 2:**

It is difficult to visually separate the different symbols and colors. In addition, grid lines would be appreciated. Further, the time series of AOD at 340 and 1020 nm could be omitted as the wavelength dependency is already illustrated by angstrom exponents.

**The reviewer is right, that the wavelength dependency is already illustrated by the angstrom exponent. Still, we want to keep the information in this plot. For better visualization we changed the color of the symbols.**

**Section 3.2**:

The introduction of this section claims that the presentation of three case studies with pure aerosol conditions is important for later studies of aerosol mixtures. However, section 1 mentions the availability of many previous field campaigns for studying pure aerosol conditions and the importance of the A-LIFE campaign for investigating mixed conditions. It is very important to add at least one extended case study with mixed conditions. The extended presentation of the mixed case should also include profiles of aerosol type and dust fraction as well as data of non-lidar measurements (AERONET, FALCON). In order to keep the manuscript short, case studies of 9 and 21 April are unnecessary.

**We do not want to include other case studies or remove case studies presented in the current version of the manuscript. We have chosen these case studies to ensure, that optical properties**

different from the know values for pure cases are really due to mixture and not due to transport or aging. A more detailed evaluation of the mixtures would be beyond the scope of this study. A detailed analysis of the in-situ measurements can be found by Teri et al., 2024.

**Figures 4-6:**

Adding a label ( a), b), c), d) ) to the different panels would be more reader-friendly. Additional panels with profiles of angstrom exponents and dust fraction need to be added. These will allow for a better comparison with AERONET data and provide a better connection with Table 2. The Quicklook could be as a top row panel above the profile panels.

**We believe that the figures are quite clear to understand and thus do not need any labeling. We also don't want to include more information like the dust fraction, as we investigate rather pure cases here. The profile of the dust fraction would not provide additional benefit. As the case studies are done for night-time measurements they are not directly comparable with the daytime AERONET measurements. Thus, giving the Angström Exponent would not allow for a better comparison. It furthermore can be derived from the extinction coefficient of both wavelengths.**

It is very advanced that the authors present systematic uncertainties of all profiles. Nevertheless, there must be statistical uncertainties, too. As such they need to be included in the plot.

**We did not include the statistical errors as those are only available for the Raman measurements and were not included in the references for the depolarization measurements. We would have to come up with a new method of error propagation for the statistical error of the depolarization measurements.**

**Section 3.4:**

Please reconsider whether it is really the altitude /signal-to-noise ratio which limits the retrieval of PLDR. In most cases, it is a too low aerosol amount making the retrieval of PLDR mathematically unstable. Thusly, it depends on atmospheric situation, not measurement setup whether PLDR can be retrieved or not.

**Actually, it is the small signal to noise ratio that limits the retrieval of the PLDR from the measured VLDR, but it is also correct that the smaller the backscatter ratio is, the higher the SNR must be to retrieve the PLDR with a given relative uncertainty. We include this in the sentence as:**

**"*As the signal to noise ratio was too small to retrieve the PLDR with sufficient accuracy for overpasses at flight altitudes >7 km with low backscatter ratios, we restrict our evaluation to the extinction coefficient in those cases.*"**

Further, it is unclear from which of the two lidar systems the reported values and findings originate. It can be assumed that this data come from POLIS system. Why are no findings from PollyXT reported in this section?

**We included in the text, that the lidar values were measured by POLIS. We also investigated if there are differences between the different lidar systems but found good agreement within the uncertainty ranges. For a better visualization of the comparison we decided to not include**

the POLLY$^{XT}$ data in Figure 7. But we extended the text to give the information about POLIS and POLLY$^{XT}$ agreement.

Again, there is the need for further explanation on how the extinction coefficients have been obtained from daytime measurements.

**For the day-time measurements the extinction coefficient is retrieved with the Fernald-Klett algorithm, as described in the Methodology section.**

**Section 3.5:**

The introduction paragraph of this section is only a replication of previous statements.

**We removed part of this introduction paragraph.**

**Section 3.5.1:**

The description of the typing algorithm should go to section 2. Thresholds used for the typing should be printed as lines in Figure 8. It would also be helpful for the reader if the individual points are color-coded by type. The highlighted daily mean values have no real benefit because the change between atmospheric situations does not follow calendar days. It would be better to highlight the measurements during FALCON flights which were used for the typing in Table 2.

**We moved the description of the typing to section 2 and extended the description. We do not want to include lines in Figure 8 to highlight the thresholds but color-coded the different types. We kept the daily mean values for the measurements used in Table 2.**

**Section 3.5.2:**

The description of aerosol typing from lidar data is in conflict with the description in section 2.4. It rather seems that the typing has been obtained from ancillary data (trajectories) and not from optical properties. How else can it be explained that the new type "Arabian dust" was identified in this study, but was not known in previous studies?

**For the identification of the Arabian dust cases ancillary data (trajectories, source investigation, satellite observations) were used. The optical properties were than investigated in detail, as also presented in a case study and in the following included into the aerosol typing using the lidar ratio difference to distinguish between Saharan dust and Arabian dust. Furthermore, we included also the investigations of former studies to come up with a threshold to distinguish between Saharan and Arabian dust. We included the following text to make that clearer.**

*'Floutsi et al. (2023) already included the separation between Saharan and Arabian dust. But they rather did a data collection than a classification. HETEAC (Wandinger et al., 2023) was made more flexible to be applied to multiwavelength observations. The resulting HETEAC-Flex (Floutsi et al., 2024 ) includes optical properties for Saharan dust separate from Arabian dust. We could check how well their optical properties are in line with the A-LIFE observations.'*

Table 1 and Figure 9 needs a much more detailed discussion. Why are there missing values in Table 1? What are the measurement times? It would also be helpful for the

comparison, if aerosol types and dust fractions derived from both lidars would be included in the table and their differences discussed.

**Missing values occur when no measurements were available at the specific times. We added this clarification in the figure caption. We also added the measurement time for which the values have been retrieved. But we do not want to include dust fractions in this table, as for our understanding it would not provide any benefit for further aerosol classification.**

**Figure 9:**

The difference between "dust mixture" and "polluted dust" needs to be explained in this paper. References to previous works is not sufficient here.

**To be more precise here, we added that 'dust mixtures' refer to mixtures of dust with marine.**

Some individual data points require more discussion. There is one "polluted marine" (25 April?) which is clearly located within the pollution cluster and one "dust mixture" (20 April?) between Saharan and Arabian dust clusters. How can those data points be so far outside their clusters if the optical properties have been used for typing?

**We had a typo in running our classification. We corrected for that.**

**Section 4.1 and Table 2:**

First of all, a translation table explaining all the different terms for aerosol types is dearly missing in this paper. Terms are different from instrument to instrument, but also from section to section (e.g. lidar "mixed pollution" and AERONET "polluted mixture" were not mentioned anywhere before Table 2). Those inaccuracies can be solved by a careful editing of the text.

**The reviewer is right, there are inaccuracies in the description of the AERONET based classification; especially with respect to the categories. We revised the AERONET classification part to be closer to the method proposed by Toledano et al. With that we reduced the number of categories to marine (not available in this study), dust, dust mixture, polluted mixture and pollution/smoke. We revised the description of the AERONET classification to be clearer.**

*'For the AERONET based aerosol typing we use the scatter plot of Ångström exponent (440-870 nm) vs. AOD at 500 nm as proposed by Toledano et al. (2009, 2011). Values of the Ångström exponent (AE) larger than ~1.2 serve as indication for smoke/pollution, independent of the AOD. Ångström exponents of <0.5 serve as indication for dust (Toledano et al., 2009, 2011, 2019) or marine aerosols. Following Toledano et al., 2011, a threshold of AOD=0.15 is used to separate dust and marine aerosols. Measurement points with AOD<0.15 and AE<0.5 are classified as marine, while measurement points with AOD>0.15 and AE<0.5 are classified as dust. Ångström exponents between 0.5 and 1.2 serve as indication for mixtures. We further subdivide this value range in dust mixtures for AE larger 0.5 a value of the AERONET derived Fine Mode Fraction larger 0.5 (AE values of ~0.8). Values with Fine Mode Fraction < 0.5 (AE values >~0.8) and AE values smaller 1.2 are classified as polluted mixture.'*

The quality and usefulness of this paper could significantly be improved by a discussion on how typing from one method (e.g. lidar) could be compared to typing from another method (e.g. in-situ). Obviously, such comparisons are difficult because instruments

measure different quantities (optical properties vs. size distribution). Nevertheless, if a direct "translation" is impossible, the authors should at least discuss the difficulties.

**Thank you for pointing this out. It is indeed difficult to directly translate the typing of from one method to another method. We mention this in the text including the following paragraph:**

*'The comparison of the different aerosol typing schemes highlight that, although the dominating aerosol type is captured quite well, it is hard to directly compare the outcome in detail. The different schemes rely on different measured quantities (e.g. optical properties vs. size distribution and microphysical properties). Thus, also the results of the aerosol classification schemes can provide a different degree of detail. It is important to carefully investigate if the chosen method provides sufficient information for specific studies they are used.'*

In general, the discussion of Table 2 is way too short. This table contains the main findings of the paper (which has "aerosol type classification" in the title) and deserves more attention. Especially since it would be interesting to learn more about the days with complex situations (5,6,11 April as mentioned in the text). Why not present one of those complex cases in the case study section?

**We extended the discussion of Table 2, as the reviewer is right, this is one of the main findings of this paper. Except for the 5 April we did not include these days as a case study, but we included a table in the supplement presenting information on the in-situ measured data of these complex days. For a more detailed analysis we refer to the publication by Teri et al. 2024 (doi.org/10.5194/egusphere-2024-701) investigating in depth the in-situ measurements and the contribution of pollution in dust mixtures.**

**Section 4.2:**

The introduction of in-situ total mass concentrations as a "referee" after discussing the differences between lidar and FLEXPART retrievals (lines 450-452) seems odd.

**We agree and changed that.**

It would be helpful to add lidar derived total mass concentrations (not only dust component) to Figure 10 for a better comparison with in-situ total mass concentration. Uncertainties due to omitting the non-dust fraction in mass estimates should at least be discussed.

**We looked at the comparison of the lidar derived total mass concentration and the in-situ derived total mass concentration. However, we found not a significant improvement of the agreement. This is due to the different volume sampled by lidar and in-situ but also due to the different assumptions considered in calculating the total mass concentration. Another reason is, that the optical properties during the dust dominated events are mainly determined by dust, while the in-situ measurements also have a good characterization of the minor contributing aerosols to be included in the calculation. We expanded like the following:**

*'The total mass concentration even exceeds the lidar derived dust mass concentration for most of those days, especially when the dust mass concentration was large. Considering also the non-dust contribution in the comparison does not result in a significant improvement of the comparison. The lidar derived optical properties during dust dominating day are mainly*

*determined by dust aerosol, while the in-situ measurements better characterize the minor contributing aerosol components which are included in the in-situ derived total mass concentration. Further differences occur due to the different averaging time of the lidar and the in-situ measurements and thus the sampled volume.'*

Would it be possible to estimate a dust fraction from in-situ data? The subclasses "pure", "moderately-polluted" and "polluted" should at least allow for a rough estimate.

**It might be possible to do so, but it is not in the scope of this study. It might be done in a follow on in-situ based study.**

Why are no Polly$^{XT}$ results included in Figure 10? Please elaborate.

**POLIS and POLLYXT data were analyzed and provided by different groups. The main lidar system used in this study is the POLIS lidar. PollyXT is used to cross-check the values of POLIS. The measurements and analysis of the two systems do not show significant differences. As we would use the same method to retrieve the dust fraction and dust mass concentration we do not expect a benefit of including the PollyXT data in Figure 10.**

**Minor points:**

L41: add comma after properties

**Done**

L53: add comma after step. A next step -> the next step

**Done**

L 179: the sentence has 2 times "obtained"

**Changed**

Figure 3: the 24h symbols are not visible.

**We revised the Figure to make the 24h symbols more visible.**

L243-244: strange wording to start a section.

**Changed**

L 246: lidar measurement show -> the lidar measurement shows

**Corrected**

Figure 7: the lower panel is too busy. It would be better to present the dust/ non-dust extinction coefficients in a separate panel. Again, statistical uncertainties are missing.

**We followed the advice of Reviewer 1 and removed the dust/non-dust extinction coefficients and just kept the dust backscatter and volume fraction.**

Table 2: again, what are statistical errors?

**The statistical error strongly depends on the averaging length. Thus, we do not think, that the values would benefit from giving them here. However, we choose the averaging length in a way that the standard deviation was small and well below the statistical uncertainties.**

Figure 9: What are the error bars? The symbols in the legend are very small and hard to see. The legend should list all aerosol types. It is confusing to have some of them listed in the legend, others in the caption. The plots would be easier to read without aerosol types which are not used in this study (like volcanic ash). This would allow to plot the existing pollution points in gray, allowing the pollution data points of this study to be plotted in black for better contrast, especially in the left panel. Another option for clarification would be not to show all individual previous data points, but only cluster boundaries as polygon lines.

**The error bars show the mean systematic error. We added that in the figure caption. We furthermore increased the legend for better visibility. However, we do not want to change the figure in a way to remove former data points.**

---

## Author Comment (AC2)

**We thank the reviewers for the careful reading and the suggestions that helped us improve our manuscript. The answers to the reviewers will be given after each comment in bold font.**

**Reply to Referee #1**

The paper is a nice addition to the literature of lidar measurements of lidar ratio and particle linear depolarization ratio of different aerosol types, with good support from comparisons with other methodologies. Interesting findings include support for a clear separation between the lidar ratios of Arabian and Saharan dust, and at least one case where the in situ suite suggested a pollution component in a dust mixture where the lidar could only detect dust. A section with dust mass and volume concentrations derived from a chain of assumptions is less interesting partly because a lack of error characterization makes it difficult to know how quantiatively useful these estimates are, yet even the qualitative comparison revealed a possible weakness with models' dust transport making another useful conclusion.

**Thanks for this valuable comment on the dust mass and volume concentration. We tried to better account for the errors and uncertainties that come along with the calculation of the dust mass and volume concentration to hopefully give more reliable data.**

While the manuscript is generally good, there are sections that seem frustratingly like a first draft. These include statements that are unsupported or vaguely approximate where precision is called for, important data presented in tables or figures without much discussion or analysis in the text, and figures with inconsistencies in presentation that make them difficult to relate to each other or with the text. Also, in many cases it is difficult to find a description of which instruments were used to derive specific data items in the figures and tables. (In most cases, I think I can infer which instruments, but can't find text to confirm my impression.) There are probably also a least a few actual errors in numbers in the text and figures, so these should all be checked thoroughly. I think it's important to improve these points and not ignore them, but I think all my suggestions should really be quite straightforward and easy for the authors to improve.

**Thank you for pointing this out. We carefully revised the manuscript.**

**Some important specific points:**

How were the lidar values that are quoted in the abstract and conclusion derived? From single measurements, or from averages over the profiles of the selected case studies, or aggregates over every point in the mission identified as a specific type? Does it combine POLIS and POLLY-XT measurements or refer only to POLIS? This should be stated clearly in whichever section is most relevant to the calculation and again in the conclusion.

**The mean values give the mean for the different types including all measurements that have been performed for the specific type and listed in Table 1. We modified the text accordingly.**

Error bars and quoted uncertainties throughout refer only to systematic errors. It's excellent that the team has characterized the systematic uncertainties so carefully, since that is sometimes neglected. However, this methodology also mentions significant

averaging, so there is random error that's not represented. There is also significant natural variability and it's important to have an estimate of the variability, to better support conclusions about whether different studies observe comparable values. It seems like multiple measurements are used to derive the quoted values of lidar ratio and PLDR (i.e. in the abstract and conclusion), so a standard deviation would be easy to calculate. (Measurement random error would probably have to come from a propagation of errors that the authors may not be prepared to do, but if it is available, this should be reported as well).

**Thank you for pointing out the necessity of error calculations. For comparability to other studies, we added the standard deviation to the mean values shown in the abstract and in the conclusion. However, the two groups providing the lidar data did not do the same completeness of error propagation. Thus, to keep the data comparable we choose to only give the systematic errors and standard deviation.**

At line 125, it's stated that the manuscript will "intercompare the measurements and resulting classification" (between POLLY_XT and POLIS). This intercomparison was not done. While both sets of measurements appear in Table 1, there's no discussion of them. Table 2 does not show classifications for the two lidar separately and although the separate classifications appear in the scatter plot of Figure 9, there's no way for a reader to identify simultaneous measurements and again no analysis or discussion. Indeed it is also difficult to tell from the text whether both or just one of the instruments' results were used in various places, such as the summary values quoted in the abstract, and for the lidar classification in Table 2.

**The reviewer is right, we did not intercompare the typing using POLIS and Polly$^{XT}$ measurements. We added a paragraph after the general typing schemes:**

**'As both lidar POLIS and PollyXT were located site by site for during the A-LIFE campaign, we can use the measurements to check if and how the analysis of different lidar systems with different algorithms done by different research groups affect the outcome. As mentioned in Section 3.4, we found now significant differences in the retrieved extensive optical properties (i.e. the extinction coefficient). With the PLDR and lidar ratio presented in Table 2 and Figure 9 we can also check if there are significant differences in the retrieved intensive optical properties. Although the mean values for the lidar ratio (both wavelengths) partly differ by 10 sr or more, considering the uncertainty range of the retrieved values we found no significant differences. For the PLDR at 532 nm we found an agreement of the mean values within 0.02 between the two instruments. For the PLDR at 355 nm the differences of the mean values are as large as 0.06 for the dust dominated day around 20 April 2017. However, considering the uncertainty range, the differences are not significant. Differences can occur from different averaging (time and height) as well as from differences in the lidar performance (e.g. signal strength).**

**Applying the classification scheme on the intensive optical properties we find good agreement of the results between the different systems and the different wavelengths. Only for the pollution case on 11 April 2017, the classification at 355 nm and 523 nm show light differences in the assigned aerosol type. The difference in the PLDR (0.12 ± 0.02 at 355 nm and 0.04 ± 0.02 at 532 nm) causes this difference. While the aerosol type was classified as polluted dust at 355 nm, it was classified as pollution/smoke at 532 nm. POLIS measurements are missing for that day to check the validity of the PollyXT classification.'**

I would like to know more about the case where in situ typing indicated a pollution component in an airmass that the lidar could not possibly type as anything other than pure dust. This was called out in the conclusions and I agree it's an interesting finding, so it deserves more attention. Specifically what chemical or other measurements in the in situ suite indicated pollution? Is there an estimate of the mass fraction or any other measurement that would support the statement that it's a minor component (as stated in the text) or alternately "moderate" (as stated in the table) (And by the way, that also needs to be made more consistent). Would this amount of pollution be expected to impact either the radiative properties or the aerosol-cloud interactions? It might be useful to highlight this case, and/or other "difficult" cases as case studies, besides the pure type cases highlighted earlier.

**The investigation on how pollution within mineral dust layers affect the optical and radiative properties of these layers goes beyond the scope of this manuscript. But as it is an important topic it is covered in an extra publication by Teri et al. (2024). In addition, we added the optical properties of these days (5, 6, 27, 29 April for Arabian dust, and 21, 22 April for Saharan dust) measured in-cabin of the Falcon aircraft in the supplement.**

**Specific points by location in the manuscript:**

*Abstract:* Please state in the abstract that the quoted error bars represent systematic error and not variability.

**Done. In addition, we added the standard deviation.**

*(Introduction)*

Line 42 or wherever seems best to the authors, please discuss whether the analysis in this study assumes the mixing is external mixing rather than internal, and any implications.

**We added this in the Methodology chapter (2.4 Aerosol typing and aerosol type separation based on lidar measurements:**

**_'Based on findings from former studies on Saharan dust (e.g. Petzold et al., 2011), we assume a two-type external mixture of mineral dust and pollution. This assumption is in good agreement with the coordinated in-situ measurements (see Section 4).'_**

Line 47. The argument for the need for typing is vague and somewhat unconvincing. It could probably be improved by being more specific. E.g. specifically how would radiation calculations use typing information and what quantitative properties could be estimated better by knowing the type of particles?

**We added a sentence why aerosol typing can help to determine the affects of the different aerosol layers. To not go to much into detail here, we added the references to prior publications on this topic:**

**_'Remote sensing data from airborne or spaceborne measurements provide information on continental and global scale. But they cannot directly derive the particles' microphysical properties or chemical composition, and thus their radiative effect and capability to act as_**

*cloud or ice nuclei. However, those properties strongly depend on the type of particle (e.g. Groß et al., 2013; Wandinger et al., 2023).'*

*(Methodology)*

Line 112. "and for the daytime measurements". Please expand on this to explain how the lidar ratio is used for the daytime measurements. The information is in the Discussion section on page 19, but the Methodology is where I think most readers would look for it first.

**Thanks for pointing this out. We moved the explanation we gave in the Discussion section to the Methodology section.**

Section 2.4 and 2.5. There are eight or so different assumed values in these two sections to make the leap from the lidar-derived particulate depolarization ratio and backscatter to dust fraction and volume and mass concentrations. Clearly this must lead to a lot of uncertainty in the output. What is the estimated uncertainty on the assumed parameters and on the calculated outputs?

**The uncertainty of the estimated dust fraction is about 10-20 %, of the estimated volume and mass concentration it is about 10-15 %. The uncertainties are given in Figure 10, and we added the uncertainties in the text.**

Line 156. The procedure for classification with AERONET measurements should be discussed in the methodology section (currently part of 3.5.1)

**We moved the description of the procedure for AERONET base classification to the methodology section.**

Line 165. How are the two different sources of dust distinguished using in situ data?

**The two dust types are mainly distinguished based on the information of the dust source region.**

Line 169 and Section 2.8. It's confusing that the "in situ" classification is stated to use FLEXPART, and then there's also a separate FLEXPART classification. If FLEXPART is really used as an integral part of what's called in situ classification, then this needs more explanation so that readers can understand what is really based on measurements and what's based on a model. If it's only to separate Saharan and Arabian dust then it would be better to have the in situ classification in table 2 only indicate "dust" and leave the FLEXPART classification as a completely separate entry.

**The airborne A-LIFE data set was classified into 4 main classes (Saharan dust, Arabian dust, mixtures with and without coarse mode) based on a threshold value of in-situ measured coarse mode number concentration and simulations with the Lagrangian particle dispersion model FLEXPART. FLEXPART results were used to decide whether a measurement was classified as a dust layer, and to distinguish between Saharan and Arabian dust. Each of the four main aerosol types was further separated into three sub-classes (clean, moderately-polluted and polluted) based on threshold values of the ratio between refractory black carbon mass and coarse mode number concentration. Details about the A-LIFE aerosol classification scheme can be found in Weinzierl et al. (2024, in prep.), and Teri et al. (2024).**

L183 "an aerosol type". The FLEXPART aerosol labels look like they are probably components that can appear simultaneously, rather than a single aerosol type per one-minute section. Please clarify. That is, in Table 2, where groups are listed, such as "SD, OM, SO4, SS", is the reader to understand that these are multiple components found together? Consider changing the wording to say something like "were assigned aerosol components" instead of "an aerosol type".

**We followed the reviewer's advice and changed that accordingly.**

*(Results)*

Line 193. Please specify which satellite measurements.

**We used a number of information amongst them, measurements of MSG. We added this in the text.**

Figure 1. It is difficult to pick out data for specific dates mentioned in the text, exacerbated by having 6 minor ticks for each 5 days, a difficult interpolation to do by eye. Can the axis please be reformed to be similar to Figure 2 where each minor tick represents one day?

**You're completely right. It is hard to spot specific days. Thank you for your comment. We updated Fig. 1 to have exactly 5 ticks for 5 days.**

Figure 1 and Figure 2. It would also be very helpful to annotate the x-axis on both figures with lines to indicate the major events and case studies that are discussed in the text.

**We modified the Figures accordingly.**

Line 210 "Similar to the Arabian dust events ... almost wavelength independent". I don't see that the Arabian dust cases can be described this way, especially the first. Please reword.

**We reworded the sentence as follows:**

**'Both events are characterized by low Angstrom exponent indicating no or low wavelength dependence. But while the Angstrom exponent (440-870 nm and 380-500 nm) during the Saharan dust event shows typical values of ~0.2 (Toledano et al., 2009; Groß et al., 2011) the Angstrom exponent (440-870 nm and 380-500 nm) during the Arabian dust event shows slightly larger values of ~0.6 for 440-870 nm and ~0.8 for 380-500 nm. Both events show a large contribution of the coarse mode particles to the overall AOD at 500 nm.'**

How does the AOD summed from the lidar extinction compare to the AERONET AOD? This could help verify that the AERONET is not impacted by clouds, highlight any potential issues from day night differences, and potentially help support the subsequent discussion about the mass concentration differences between the lidar calculations and AERONET.

**We investigated the agreement of lidar and sun photometer derived AOD for some of the coordinated measurements and found good agreement. However, this cannot verify that the AERONET algorithm filters out all contribution of clouds, as the AERONET and lidar measurements are different in resolution and sampled volume. We do not include the AERONET measurements in the discussion about mass concentration.**

Line 232 What's the justification for calling these cases "pure"? The in situ classification in the table does not seem to support calling them all pure.

**The reviewer is right, the in-situ measurements indicated enhanced coarse mode particles in during these events but still indicated that the major aerosol type was pollution aerosol. We rephrased the sentence to 'with a dominance of'**

Figure 3. While the backtrajectories show the direction the airmasses came from, they do not seem to go far enough back to show the sources. Both the pollution and Saharan dust tracks are just as dispersed in altitude at the earliest time shown on the track as at the measurement location. Wouldn't we expect to see tracks converging in altitude to the injection height at the source? Later, at line 280, it says "(Figure 3) indicated the western Saharan region" but that's not actually true, because the trajectories don't go far enough back to reach the western part of the Sahara.

**This is right, the back trajectories shown in Figure 3 shall highlight that in fact the airmasses are coming from different direction. For a detailed source analysis, we used the FLEXPART analysis.**

Line 244. Again, which satellites?

**Amongst others MSG and MODIS**

Figure 2 shows two different Angstrom exponents. The text frequently refers to the Angstrom exponent, but doesn't say which one.

**We added the wavelength pairs we used for the Angstrom exponents.**

Line 248. How were the PLDR values 0.24 +- 0.02 and 0.27 +- 0.01 arrived at? They really do not seem to be averages of the values displayed in Figure 4 between 0.5 and 2.0 km altitude, as implied. Some subset near the top of that range may average to 0.27 at 532, but there doesn't seem to be any single value above 0.23 at 355, so a subsetted range can't really explain the discrepancy there. Are these correct?

**The reviewer is right, for 355 nm this was a typo. The right value is 0.21+/- 0.02. We corrected this typo.**

Line 273. Similarly I can't understand how the values at 355 and 532 differ by 0.01 when the profiles in Figure 5 seem to completely overlap. Is it because the 532 nm profile goes up a bin or two higher in altitude than 355 where the values are largest? It also seems strange that it wouldn't be acknowledged in the text how similar these two lines are. Is there any error here?

**We checked our analysis and the averaging range for both wavelength and found no change in these values.**

Figure 6 has a clear error in how the PLDR error bars are plotted. Both green and blue error bars are plotted around the green line with none around the blue line. It's also remarkable how much less vertical variability the green line has. Is there any difference in the smoothing/resolution between these?

**Thanks for pointing this out. We corrected the figure. No, there are no differences in smoothing or resolution.**

The quantities quoted in the case studies seem to be somewhat sloppy estimates. For instance, the 380-500 Angstrom exponents on April 5 on Figure 2 cover the range from about 0.8 to 1.3 and are described as "characterized by low Angstrom exponent of about 0.5" (line 246). Even if its the long wavelength Angstrom exponent that's meant, 0.5 seems to be just a bit lower than any of the values shown on that day. The 355 nm lidar ratio is described as "wavelength independent" (line 250) but it is not specified that this applies only to POLIS and not to POLLY_XT.

**We changed the description of the case studies and also added that for the detailed case studies only the POLIS measurements were used.**

Line 267 describes the AERONET AOD as "about 0.15 at 500" and Angstrom of "about 1.5" but the AOD never gets to 0.15 on that day and seems to vary around about 0.12, and 1.5 is at the high end of the variability of the Angstrom exponent. Instead of reading "about" this or that, I would rather see a quantitative average and standard deviation over each day (or whatever period is desired) for a more accurate discussion. Same for the discussion of the lidar values which are also described as ">" or "around" (Line 271) instead of quantitatively.

**We modified this paragraph to give a more quantitative description.**

*'For the AERONET based aerosol typing we use the scatter plot of Ångström exponent (440-870 nm) vs. AOD at 500 nm as proposed by Toledano et al. (2009, 2011). Values of the Ångström exponent (AE) larger than ~1.2 serve as indication for smoke/pollution, independent of the AOD. Ångström exponents of <0.5 serve as indication for dust (Toledano et al., 2009, 2011, 2019) or marine aerosols. Following Toledano et al., 2011, a threshold of AOD=0.15 is used to separate dust and marine aerosols. Measurement points with AOD<0.15 and AE<0.5 are classified as marine, while measurement points with AOD>0.15 and AE<0.5 are classified as dust. Ångström exponents between 0.5 and 1.2 serve as indication for mixtures. We further subdivide this value range in dust mixtures for AE larger 0.5 a value of the AERONET derived Fine Mode Fraction larger 0.5 (AE values of ~0.8). Values with Fine Mode Fraction < 0.5 (AE values >~0.8) and AE values smaller 1.2 are classified as polluted mixture.'*

Line 286 "Those values have been reported recently". That is an odd way to phrase this unless the same exact values were observed in another study. I assume the authors mean that the current values are similar or consistent with previous studies. Please specify the values from the other studies so readers can see this for themselves. This applies in other parts of the manuscript also. Please always explicitly quote the values that are being compared.

**We followed the reviewer's advice and changed this phrase to '*Similar values for Saharan dust after a transport of several days were also reported from recent studies*'.**

Line 296.  I don't fully understand how extinction can be retrieved in the daytime but lidar ratio can't. My understanding is that the extinction retrieval itself is quite difficult if the signal is noisy, but I believe the backscatter retrieval is not as sensitive (is that correct?) so it would seem that if conditions are sufficient to retrieve extinction, lidar ratio would not be much harder. I understand this is probably explained fully in prior papers by the research group, but a summary of the explanation for this in the methodology section would be appreciated.

**For the day-time measurements the extinction coefficient is retrieved with the Fernald-Klett algorithm, as described in the Methodology section. As described, this algorithm uses the lidar ratio as an input.**

Line 298.  It's not really the measurements' signal-to-noise ratio that prevents the retrieval of PLDR at high altitudes, is it?  That is, the measurements probably have sufficient SNR to retrieve a reasonably precise volume depolarization ratio, but the low values of the scattering ratio prevent converting it to PLDR with reliable accuracy becasue of the singularity in the conversion equation.

**Actually, it is the small signal to noise ratio that limits the retrieval of the PLDR from the measured VLDR, but it is also correct that the smaller the backscatter ratio is, the higher the SNR must be to retrieve the PLDR with a given relative uncertainty. We include this in the sentence as:**

**"*As the signal to noise ratio was too small to retrieve the PLDR with sufficient accuracy for overpasses at flight altitudes >7 km with low backscatter ratios, we restrict our evaluation to the extinction coefficient in those cases.*"**

The switch from time (in Figures 1 and 2) to overpass number (in Figures 7) is not very user-friendly. Would it help to plot Figure 7 against time like Figs 1 and 2 (even though the symbols will not be linearly spaced)?  Even if not, please consider annotating the x axis with lines indicating the dates of the major aerosol events identified in the text of the paper to help orient the reader to the new x-axis.  A callout to the appendix in the figure caption might also help slightly.

**Although we think, that the figure becomes less well structured and the comparison of in-situ and lidar measurements less clear, we agree that the change to overpass numbers is not very user-friendly. Thus, we followed the advice of the reviewer to change the x-axis to stay with the formatting of the former figures.**

The lowest panel of Figure 7 is very busy and hard to read with so many different symbols, especially with many missing data points.  Maybe consider moving the dust and non-dust extinction up to the PLDR plot on a second y axis.  That would also have the benefit of making it easier to compare dust fraction to PLDR.  Indeed it might be more informative to show it as dust fraction rather than as separate dust and non-dust components.  In fact, the PLDR and dust fraction data could very well be a separate Figure 8, since they have nothing to do with the in situ comparison, as far as I can see, and separating them would make them easier to understand.

**We agree that the dust fraction and the PLDR has nothing to do with the in-situ comparison. Thus, we follow the reviewer's advice to separate the comparison from the dust contribution. As we already investigate the dust contribution with respect to the dust fraction on the**

**backscatter coefficient and the volume, we decided to simply remove the dust AOD and non-dust AOD in Figure 7.**

Indeed, could there be an intercomparison of dust fraction with in situ?  In situ has a particle size distribution and coarse mode fraction, so is there any comparison available with some version of the dust fraction derived from PLDR?

**To have such an intercomparison is beyond the focus of this publication, but the in-situ group is working on such activities.**

Line 303. "In the lowermost layers, the values range..."  Why is there no analysis or discussion of the agreement in these more important layers?  There is sometimes disagreement beyond the error bars.  This is probably not of much concern, but since there was a stated intent to compare the two, a comparison should be discussed both when there is agreement and when there isn't.

**We added a discussion on the disagreements of the in-situ and the lidar derived values:**

*'Although the lidar and in-situ derived extinction coefficients show the same behaviour, differences between both methods are obvious on dust dominated days, with the in-situ values exceeding the lidar derived extinction coefficients. The largest differences we found was about 0.05 km$^{-1}$ within the Saharan dust layer. The differences result partly from the different methods used to derive the extinction coefficient and partly from the different volume sampled by in-situ and lidar measurements.'*

Line 304. "The lidar ratio ... helps to distinguish". This is either a non-sequitor or a typo since the sentences before this talk about extinction, and after it talk about PLDR.

**The reviewer is right, this was a typo and is corrected now.**

Line 306.  Include some discussion of what happens when the PLDR is 0.32 while 0.3 was assumed as the value for pure dust. It looks like maybe no dust contribution is reported in that case (such as overpass #30)

**We mentioned this topic in Section 2.4: *'It has to be considered, that a deviation of the actual measured PLDR and the one used for the type separation can lead to and over or underestimation of the contribution of the two assumed aerosol types. And thus, it is important to investigate the uncertainties of the type separation, including also uncertainties of the input values.'*, and discussed the resulting overestimation along with Figure 10: *'As we find a deviation of the measured value of the PLDR of 0.32 on 27 April 2017 and the input value used for type separation of 0.3 the retrieved dust fraction to the bsc and the volume is overestimated, leading to mean values slightly larger than 1.'***

Figure 7 caption. Which instrument are the dust and non-dust contributions from?

**We added that those values were derived from the lidar analysis.**

Figure 7.  I believe the dust and non-dust contributions are derived via the PLDR and backscatter from the POLIS instrument.  Why don't the dust and non-dust extinctions add up to the total 532 nm POLIS extinction?  For instance, the first two low-altitude overpasses (#1 and #3) have very similar non-dust extinction (gray) and very different

dust extinction (orange), but very similar total extinction (green) (and in fact the one with larger dust component has smaller total).

**Right, the dust and non-dust extinction coefficients were derived from POLIS measurements. Having only assumptions for the input parameters to calculate the dust and non-dust contribution (PLDR and LR for dust and non-dust component) can lead to uncertainties that, summing up the two values, lead to differences compared to the directly measured extinction coefficient. Following the suggestion of this reviewer, we removed the dust and non-dust extinction coefficient in this comparison.**

Figure 7.  How does overpass #32 have a larger dust extinction than total 532 nm extinction and how should this be interpreted?

**Please see previous comment.**

Lines 329-335.  In the section describing how aerosol types were derived from AERONET, it's somewhat difficult to understand which parts are quoting observations of the Toledano et al. references and which are talking about the methodology for typing in this particular study.  This confusion is exacerbated by the use of "about" preceeding almost every number in the paragraph.  This makes the numbers sound very vague whereas this paragraph needs to describe what discrete thresholds were actually used in the study to set the AERONET types in Table 2.  I hope the paragraph can be revised to be more clear and quantitative.

**We significantly revised the description of the AERONET classification to be clearer.**

**'For the AERONET based aerosol typing we use the scatter plot of Ångström exponent (440-870 nm) vs. AOD at 500 nm as proposed by Toledano et al. (2009, 2011). Values of the Ångström exponent (AE) larger than ~1.2 serve as indication for smoke/pollution, independent of the AOD. Ångström exponents of <0.5 serve as indication for dust (Toledano et al., 2009, 2011, 2019) or marine aerosols. Following Toledano et al., 2011, a threshold of AOD=0.15 is used to separate dust and marine aerosols. Measurement points with AOD<0.15 and AE<0.5 are classified as marine, while measurement points with AOD>0.15 and AE<0.5 are classified as dust. Ångström exponents between 0.5 and 1.2 serve as indication for mixtures. We further subdivide this value range in dust mixtures for AE larger 0.5 a value of the AERONET derived Fine Mode Fraction larger 0.5 (AE values of ~0.8). Values with Fine Mode Fraction < 0.5 (AE values >~0.8) and AE values smaller 1.2 are classified as polluted mixture.'**

Figure 8. It would also be good if the thresholds for distinguishing types were shown as lines on the left panel.

**We color-coded the different measurement points according to the type classification scheme.**

Figure 8. It would probably help the reader if the points in the scatter plot were color-coded in some way that would help pick out specific events.  Perhaps color code all of the points by date, or maybe color code only the points that correspond to the key aerosol episodes that are highlighted in this study (the Arabian and Saharan dust events and the pure pollution events).

**We followed (partly) the advice of Reviewer 2 and removed the daily mean values. We kept only those points that were used in Table 2.**

Section 3.5.2. Strangely, it's really not clear how the lidar aerosol type classifications are done, despite it being the main theme of the paper. I think but am not completely sure, that they were done using only the lidar ratio and PLDR, (and not relying on backtrajectories or other information, because that would make it impossible to use those for verification). Is that correct? If so, what are the thresholds? (Or if the method is more complex than simple thresholds, describe it here). Please also show the thresholds in Figure 9. Which lidar instrument is used for the classifications in Figure 9 and Table 2? It seems like perhaps both are included in Figure 9, but the table has only one column for "lidar".

**The lidar based aerosol typing uses PLDR and lidar ratio as described in Section 2.4 Aerosol typing and aerosol type separation based on lidar measurements. A detailed description of the method including a figure with the decision tree and threshold can be found in the publications by Groß et al., 2013 and Groß et al., 2015. As the method is available and the information easily accessible we do not think that it is necessary to repeat it here.**

**Data from both instruments are shown in Figure 9, indicated by two different symbols. This is also stated in the Figure caption. For Table 2 we did not separate the results from the two lidar systems as (except for one value) both agree in their typing. Following the suggestion of this reviewer we added a more detailed description comparing the results of the two lidar systems, highlighting their good agreement. We added a note, that both lidar systems agree in the lidar based classification for Table 2.**

Table 1. Extinction should be included in the table, since it's also shown in Figures and discussed in the text.

**As the properties given in Table 1 are the values used for aerosol typing, we do not want to include the extinction coefficient, as it is not a property used for aerosol typing.**

Table 1 caption. "For this evaluation" What exactly does that refer to? (I don't think it can mean the whole manuscript, since POLLY-XT does appear in figures.)

**We changed the caption to make clear that the values given in the table were derived from nighttime measurements and removed POLIS, as also the Polly$^{xt}$ night time measurements were used.**

Figure 9. The blue (polluted marine) point at 60 sr looks rather odd. It seems like it is between points labeled pollution on both the lidar ratio and depolarization ratio axes. If this is real, I would like to read an explanation or discussion of this point (If the data are being classified by the two lidar quantities using thresholds, without using other non-lidar data, I don't see how this point and the black pollution point at about 0.12 PLDR could both be as they are shown. Maybe one is an error, or maybe I misunderstand the method. Either way requires a revision, even if it's just to make the explanation of the method more explicit.)

**We corrected Figure 9.**

*(Discussion)*

Line 400. "verified by the evaluation of air mass source regions". If the back-trajectories are used in both the "in situ" classification and the "lidar" classification, doesn't that complicate the ability to usefully compare them to each other?

**We did not use back-trajectories in the lidar classification.**

Line 410-411. (repeat from above) I would like to know more about this more difficult case where in situ typing indicated a pollution component in an airmass that the lidar could not possibly type as anything other than pure dust. Specifically what chemical or other measurements in the in situ suite indicated pollution? Is there an estimate of the mass fraction or any other measurement that would support the statement that it's a minor component (as stated in the text) or alternately "moderate" (as stated in the table) (And by the way, that also needs to be made more consistent). Would this amount of pollution be expected to impact either the radiative properties or the aerosol-cloud interactions? This could be an important conclusion, that depolarization indicating pure dust could still include measureable amounts of pollution, especially if it would be expected to have relevant impacts. It might be useful to highlight this case, and/or other "difficult" cases as case studies, besides the pure type cases highlighted earlier.

**There is a separate publication by Teri et al. (doi.org/10.5194/egusphere-2024-701) handling this topic. We refer to this publication in our study. Especially in the supplement of this publication, the requested information can be found.**

**We corrected our expression about 'minor' and 'moderate' to be consistent.**

Table 2. AERONET classifications include a surprising number of similar sounding categories, "Dust and pollution", "mixed dust", "polluted mixture". What is the difference between these, both qualitatively and quantitatively?

**The reviewer is right, there are a number of similar categories which actually do not reflect exactly the classification by Toledano et al. but were driven by the a priori information from source investigation and in-situ measurements. We revised the AERONET classification part to be closer to the method proposed by Toledano et al. and with that independent from a priori information influencing the classification. With that we reduced the number of categories to marine (not available in this study), dust, dust mixture, polluted mixture and pollution/smoke. We revised the description of the AERONET classification to be clearer.**

Table 2. AERONET 11 April. I'm surprised by "mixed dust". In Fig 2, this date has some of the lowest AE. Wouldn't that make it unambiguously pollution based on how the AERONET classification was described? (As I've said already, I don't find the desription of the method very clear, so I could very well be wrong, but would like to understand better).

**The low AE in the AERONET measurements point towards larger particle sizes (likely dust) but the mean AE is just above the threshold of 0.5 to be classified as Dust Mixture instead of dust.**

Table 2. Lidar column. Which lidar's classification is this?

**Please see prior comments**

Table 2. Lidar column. "Mixed pollution" (11 April and 25 April) is not a category in Fig 9. What category is this meant to correspond to? Please make the category labels consistent across all figures and tables.

**Thanks for pointing this out. We corrected it.**

Table 2. "mixed pollution" 25 April. Why would this be "mixed pollution" rather than just pollution, where it seems to fall on Figure 9? (if the values are 0.03 PLDR /60 sr at 355 nm, as I think they are)?

**This is right, and we corrected it.**

Table 2. FLEXPART 25 April. What is the meaning of "SS(dust)"?

**The sea salt (SS) contribution was independent from '(dust)'. There was a chance for some minor dust contribution. However, the way we indicated it in the Table was unclear. Thus, we corrected it.**

Line 450. "while FLEXPART estimates only about 30 ug/m3". What about dust mass fraction or total mass? That is, does FLEXPART agree better with the total mass, but disagree that the mass is mostly dust, or have a similar fraction but disagree with the total (or neither)?

**We checked the total mass of FLEXPART but it does not show a better agreement. Thus, we still assume that FLEXPART underestimates the dust mass.**

Figure 10. What is the interpretation if the volume fraction is bigger than bsc fraction, or the reverse, or when there is a large difference? More specifically, how is it possible to have a case like overpass #10 where the volume fraction is apparently zero but the backscatter fraction is near 60%?

**Looking at the uncertainty range in the retrieved bsc fraction and volume fraction, we think it would be going to far to interpret differences in the volume fraction and bsc fraction. To highlight that, we included the uncertainties in Figure 10, showing that the differences are within the uncertainties. We also added some text to make that clearer.**

*'Both dust fractions agree well within the uncertainty ranges, except for 11 April 2017 when the dust fraction to the bsc significantly exceeds the volume dust fraction. This difference might be caused by a wrong assumption of the contributing types, and thus of the chosen conversion factor.'*

*(Summary and Conclusion)*

Line 464-478 The summary and conclusion section has a confusing flow (especially the first long paragraph). It would be good to reorganize it to group similar thoughts together. It might be useful to have a larger number of distinct paragraphs.

**We restructured the Summary and Conclusion with specific focus on the first paragraph.**

Line 465 "compared to the lidar ratio found for Saharan dust". This would preferably be followed immediately by giving the the lidar ratios for Saharan dust.

**We added the values for Saharan dust.**

Line 467-469. Please put the mean values of the PLDR of the measured Arabian dust immediately after mentioning it (at the beginning of the sentence). As it is, it reads as if the quoted values are the values "for Saharan mineral dust close to the source region".

**We followed the reviewer's advice.**

Line 469. There also seems to be a typo, since the numbers for 355 and 532 are the same here, but different in the abstract.

**The sentence was misleading. We changed that by also following the reviewer's former advices.**

Line 470 and 471. "their PLDR of" and "their lidar ratio". This sentence is also constructed ambiguously, with the grammar suggesting that these refer to pollution, or perhaps to the Saharan AND Arabian layers. Only by cross-referencing with the abstract can I see these are the values for the Saharan observation.

**See previous answer. We followed the reviewer's advice and changed it.**

Line 471-473 and Line 467. These sentences about agreement with previous studies should be placed together or combined. These two very similar ideas placed far apart is part of what makes the flow hard to follow.

**We moved the later sentence to directly follow the first one.**

Line 474 "0.05" at 532 nm. The abstract says 0.04.

**We corrected that.**

Line 481-482. "The derived volume fraction of the dust... showed a lower contribution to the total volume compared to its contribution to the optical properties". I don't understand this conclusion. Perhaps spelling out what optical properties would help, but I think it is probably referring to the lidar-derived extinction and PLDR. Since the derived volume fraction is assumed from a simple formula applied to the optical properties, is this just saying that the relationship is a curve rather than proportional? (not a new discovery).

**The optical properties are just mentioned in the prior sentence.**

Line 487. "Although it could predict the dust transport in general". Was this discussed in the Discussion section?

**No, we did not include a special discussion on dust transport. We believe that this is beyond the scope of this paper.**

Line 486-488. "Models generally assume". This is the first appearance of this idea; it should be in the Discussion section.

**We followed the reviewer's advice and added this also in the discussion.**

Line 491-492. "in-situ derived total mass concentration exceeded". In the Discussion a partial explanation of this was given, that it reflects different sampling resolutions. It would be good to repeat that in the conclusion.

**Thanks for this helpful advice. We repeated the explanation here.**

Minor points:

Line 40 replace "process" with "processes"

**Done**

Line 41 replace the comma between optical and microphysical with "and"

**Done**

Line 205 "indicated by the greenish to reddish colors": please indicate which panel is being referred to.

**Done**

There should be a callout to Fig 2, probably somewhere between lines 200 and 210

**We included that.**

Line 217 replace "neglectable" with "negligible"

**Done**

Line 298.  Probably intends too small rather than too large.

**Corrected**

Line 304.  Please split this paragraph into two or more, since it covers a number of different topics.  I suggest splitting at line 304 after "campaign".

**Done**

Line 309.  Consider including the reference to the Tesche et al. paper(s) that give the methodology for deriving the dust contribution here again.

**All the references to the methodology are included in the methodology section. This includes of course the publication by Tesche et al.**

Figure 9 color legend.  Please make the dots in the legend bigger.  It is very difficult to distinguish the colors.  The text in the legend is a bit on the small side too.

**We changed that.**

Table 2 and Table 1. Include overpass number as one of the columns, to make it easier to compare with Figure 7.

**That is a very good suggestion. We added the overpass number in Table 2. However, we did not add it in Table 1 as these values are independent of the overpasses and refer to nighttime measurement.**

Table 2 caption, describe why some rows have no lidar or FLEXPART classifications.

**For those layers no lidar classification and FLEXPART classification was possible. We added this information in the figure caption. We still keep these height ranges as it completes the comparison with the AERONET classification.**

Table 2 and Figure 9 (possibly elsewhere too). "Dust mixture" in Fig 9 is "Dust mixture (marine)" in Table 2.  The position on Fig 9 does seem to suggest this is meant as exclusively a marine + dust mixture.  If so, consider changing the name to reflect the more specific meaning.

**Thanks for pointing this out. We changed it.**

Figure 10. blue "BSC" dot missing at overpass #30.  Why?

**The value was out of the range shown in the Figure. We corrected that.**

Line 477. "the different classification schemes".  I suggest spelling out what these are, to help the summary section stand by itself.

**We mentioned the methods in the prior sentence, but to be clearer, we changed the phrase to 'We compared the findings of the lidar based classification to aerosol typing based on in-situ measurements, sun-photometer, and FLEXPART transport simulations. The different classification schemes showed a very good agreement, although the sun-photometer based classification can only give a column integrated value.'.**